# Exposure to circadian disrupting environment and high-fat diet during pregnancy and lactation alter reproductive competence and lipid profiles of liver, mammary, plasma and milk of ICR mice

**Leriana Garcia Reis**[1], **Kelsey Teeple**[1], **Michayla Dinn**[1], **Jenna Schoonmaker**[1], **Sara Brook Scinto**[1], **Christina Ramires Ferreira**[2], **Theresa Casey**[1]*

1 Department of Animal Sciences, Purdue University, West Lafayette, Indiana, United States of America,
2 Bindley Bioscience Center, Purdue University, West Lafayette, Indiana, United States of America

* theresa-casey@purdue.edu

## Abstract

This study's objective was to determine the effects of pre-pregnancy obesity induced by a high-fat diet and exposure to circadian-disrupting light-dark phase shifts on birth littler size, pup survival to 24h and growth to lactation day 12, and their relationship to maternal feeding patterns, fecal corticosterone levels, milk composition, and lipid profiles of liver, plasma, mammary gland, and milk. A 2 by 2 factorial designed experiment of female ICR mice assigned to control (CON; 10% fat) or high-fat (HF; 60% fat) and either a 12-hour light-dark (LD) cycle or a chronic jet lag model of 6-hour phase-shifts (PS) in light-dark cycle every 3 days throughout pregnancy and lactation, resulted in 4 treatment groups: CON-LD, CON-PS, HF-LD and HF-PS. HF diet increased maternal pre-pregnancy body weight and elevated milk lactose. Whereas PS reduced milk lactose within the CON diet group, and increased maternal feed intake and fecal corticosterone levels. PS exposure also affected the time of day of birth. Neither PS nor HF affected birth litter size or pup survival. Only diet impacted final litter weight, with HF greater than CON. Among the 1204 lipids detected by multiple reaction monitoring (MRM)-profiling, diet altered 67.1% in milk, 58.1% in mammary gland, 27.2% in the liver, and 10.9% in plasma, with HF increasing the carbon length of diacylglycerols in the liver and milk, and carbon length of triacylglycerols in plasma, mammary gland and milk. Although exposure to PS had no overall impact on maternal lipid profiles, interactions (P < 0.05) were found between PS and diets in the phosphatidylcholine and phosphatidylethanolanine class of lipids. Findings support that high fat diet and exposure to circadian disrupting environments impact maternal feeding behavior and stress responses as well as lipid profiles, which may relate to their negative association with maternal health and offspring development.

**Data availability statement:** All relevant data are within the manuscript and its Supporting Information files.

**Funding:** The authors would like to thank the São Paulo Research Foundation (FAPESP) by the financial support given to Leriana Garcia Reis (Grant nº 2021/12819-0). The funders had no role in study design, data collection and analysis, decision to publish, or preparation of the manuscript. https://fapesp.br/en.

**Competing interests:** The authors have declared that no competing interests exist.

## Introduction

Obesity and circadian disrupting environments such rotating shift work, transmeridian travel, and behaviors that cause social jetlag are common in modern societies [1]. Epidemiological evidence links obesity and circadian disruption to the development of chronic non-communicable diseases like metabolic syndrome and cancer [2,3]. Obesity and exposure to circadian disrupting environments are also linked to reproductive dysfunction, subfertility, and poorer breastfeeding outcomes in women [4–6]. The integrated and reciprocal regulation of the circadian and metabolic systems likely confound the influence of obesity and circadian disruption on the reproductive competence of women. Both are related to development of insulin resistance and dyslipidemia and alter hormonal milieu, all of which are linked to decreased fertility and poorer lactation outcomes in humans [5].

Circadian rhythms are 24 h cycles of behavior and physiology that function to synchronize physiological processes across the body and with the external environment. Circadian clocks that generate these rhythms play a central role in maintaining metabolic homeostasis and coordinating the timing of reproductive processes like ovulation and parturition [7–10]. The 24 h rhythms generated by circadian clocks are integrated into homeostatic feedback loops and repair pathways. Responsiveness of the master clock in the suprachiasmatic nuclei (SCN) to light and peripheral clocks to feeding time enables synchronization and adaptation to seasonal changes in day length and food availability. However, exogenous and endogenous factors that function as inputs to the circadian clocks disrupt their timing and the order of physiologic functions when applied at unusual and inappropriate times and are referred to as chronodisruptors [11]. Changes in the natural light-dark cycle perturb the circadian system. Other chronodisrupters include inappropriately timed food intake and physical activity and biological stress [9–11]. Nutritional quality and obesity also play a role in maintaining robustness of the circadian system.

Orchestrated changes occur in maternal behavior and physiology to support the growth and development of the fetus during gestation and the nutritional demands of neonates during lactation. Maternal nutritional state and the external environment can impact these adaptions to pregnancy and lactation [12]. Since maternal environment governs offspring environment during pregnancy and lactation, these alterations can have profound and long-term effects on offspring development and health. For example, exposure of pregnant ewes to a changing shift work model altered maternal metabolism and increased gestation length and decreased birth weight of offspring [13]. Studies of rats found maternal exposure to chronic light-phase shifts during gestation and the first week postnatal resulted in rat offspring displaying age and gender dependent hyperleptinemia, hyperinsulinemia, poor glucose tolerance, insulin resistance and increased adiposity [14]. Exposure of offspring to maternal circadian disruption during lactation alone reduced adult body mass, increased social avoidance and hyperactivity. Whereas exclusive *in utero* exposure to maternal circadian disruption was associated with social avoidance and hyperactivity [15]. Epidemiologic data and controlled animal studies demonstrate that early exposure to maternal high-fat diet (HFD) during pregnancy and lactation also has long-term effects on offspring health, which include development of metabolic syndrome, hypertension, type 2 diabetes, obesity, cardiovascular dysfunction and alterations in neurological development [16–20].

Our overall goal is to determine if and how circadian disruption and obesity interact to affect maternal reproductive competence and the consequence to offspring. Our previous studies found that feeding a high-fat diet for 4 weeks prior to mating significantly increased body weight and percent body fat of female mice as well as altered their circadian feeding patterns, attenuated corticosterone circadian rhythms and significantly increased hair

corticosterone levels, independent of final body weight [21]. Here we describe a 2 by 2 factorial designed study to analyze the effect of circadian disruption through exposure to continuous 6h light-dark phase shifts, which affect circadian behavior, hormonal release and clock gene rhythms across multiple organs [22–25], and high-fat diet induced obesity during gestation and lactation on birth litter size and postnatal weight to lactation day 12. These findings were related to the effect of treatment and diet on maternal diurnal and nocturnal feed intake and fecal corticosterone output across pregnancy and lactation, circulating prolactin, triacylglycerol (TG) and Hb1Ac levels, an indicator of glycemic control, and milk macronutrient composition. In addition, our previous study found that high-fat diet increased the levels of malondialdehyde (MDA) in milk. MDA is a marker if oxidative damage of lipids via peroxidation. Nutrition affects lipid metabolism, and circadian clocks regulate lipid metabolism in the liver and mammary gland [26–29]. High-fat diet and circadian disruption increase oxidative stress and dyslipidemia [30,31], and thus we conducted exploratory lipidome analysis using multiple reaction monitoring (MRM) to determine if treatment and diet affected liver, plasma, mammary gland and milk lipid profiles.

## Materials and methods

### Animals and experimental design

Animal use protocols were reviewed and approved by the Institutional Animal Care and Use Committee (IACUC; protocol # 2104002135) of Purdue University prior to beginning studies. Three-week-old female ICR mice (n = 40) were obtained from Evigno (CD1, Indianapolis, IN, USA), ear tagged, acclimated for two weeks, and then randomly assigned into one of two dietary treatments: control (CON, n = 20) and high-fat (HF, n = 20). Prior to mating, the females were housed in groups of 5 based on treatment for 4 weeks and fed either a control diet (Research Diets D12450J, 10% of total kcal energy is fat and 7% of total kcal is sucrose; 3.85 kcal/g) or high-fat diet (Research Diets D12492, 60% of kcal energy is fat and 7% of kcal energy is sucrose; 5.24 kcal/g), as we have described [21]. During acclimation and the following four-week period all mice were exposed to 12h light-12h dark cycles.

After the 4-weeks on experimental diets, females were weighed (pre-pregnancy weight) and mated with males at a 2 to 1 ratio. Females were checked twice daily at 12h intervals for the presence of a vaginal plug, indicating successful mating and considered gestation day 1. Upon observation of vaginal plug or after 5 days with males, females were moved into one of the experimental light rooms where they were singularly housed and remained until 12 days postpartum. The experimental lighting was control (LD), or phase shifted (PS). The LD mice were exposed to regular 12h light (L) - 12h dark (D) cycles, with the light phase averaging 114 lux at mouse eye level. The PS mice were exposed to the same lux of light for 12h each day, and 12h of darkness, but every three days the start of the light cycle was shifted forward 6h (Fig 1). Although mice were checked twice daily for vaginal plugs after placement with males for mating, the presence was only observed in 18 of the 40 mice and there were not enough animals in each treatment group to analyze the effect of diet and light on length of gestation. The 2 X 2 factorial study design resulted in 4 treatments of CON-LD (n = 8), CON-PS (n = 9), HF-LD (n = 8) and HF-PS (n = 7). The conception rate was 85% for CON and 75% for HF.

Females were allowed to birth naturally, time of day of birth was recorded and at 24h postnatal, litters were standardized to n = 8 neonates/dam. Birth litter size, and pup survival to 24h were recorded. Litter weight was taken every 2 days from day 0 to day 12 of lactation, dams and pups were separated at 0600. Pups were weighed, and then euthanized by decapitation. After 3h of separation from pups, dams were milked. Immediately after milking, dams were euthanized via slow fill $CO_2$ inhalation, followed by cervical dislocation as a secondary

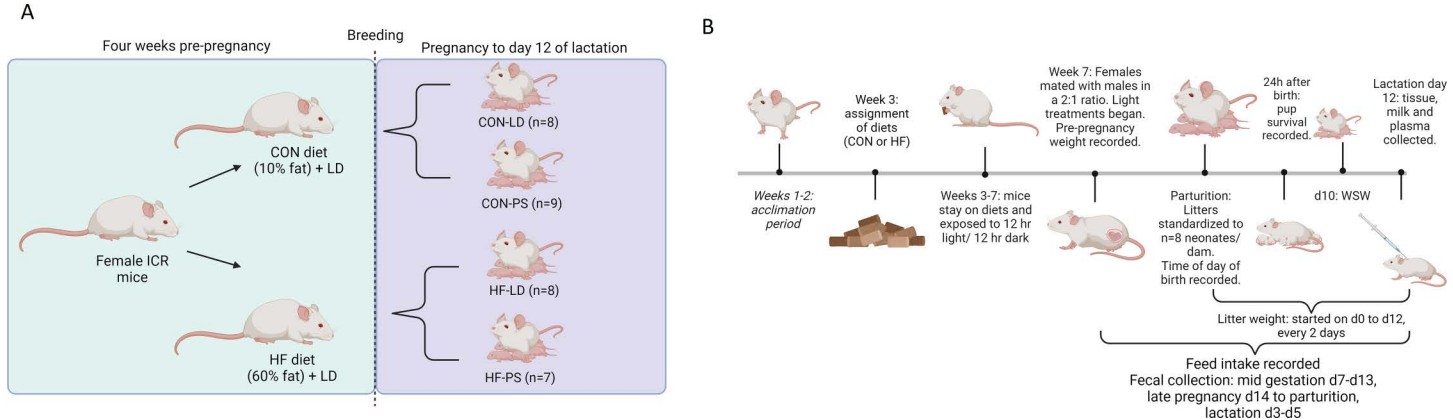

**Fig 1. Experimental design and data collection overview of the 2 x 2 factorial study.** The study investigated the combined effects of diet, high fat (60% fat, HF) or control (10% fat, CON), and light exposures that maintain homeostasis, regular 24 h cycles of 12 h of light and 12 h of dark (LD), or disrupt the circadian timing system, 6 h light phase shifts every 3 days (PS), on birth litter size, pup survival to 24 h and growth to lactation day 12, and their relationship to maternal feeding patterns, fecal corticosterone levels, milk composition, and lipid profiles of liver, plasma, mammary gland, and milk. Female ICR mice began experimental diets 4 weeks (CON or HF) prior to mating to males at a 2:1 ratio to achieve differential body weight prior to pregnancy. After mating, females remained on respective diets and were housed individually under experimental lighting conditions until 12 days postpartum. Lighting conditions included a control (LD) 12 h light (114 lux at mouse eye level) - 12 h dark cycle (LD) or a chronic 6 h light phase-shift (PS), where the light phase was advanced by 6 h every three days. This resulted in 4 treatment groups: CON-LD, CON-PS, HF-LD, and HF-PS, with final group sizes of n = 8, n = 9, n = 8, and n = 7, respectively (A). Feed intake was measured twice daily to capture 'day' (0600-1800) and 'night' (1800-0600) feeding patterns. Feces were also collected twice daily from the cages to capture day (0600-1800) and night (1800-0600) output in early pregnancy (d5-d7), late pregnancy (d14 to parturition (~d18), and lactation (d4 to d6 of lactation). Females were allowed to birth naturally, time of day of birth was recorded and at 24 h postnatal, litters were standardized to n = 8 neonates/dam. Birth litter size, and pup survival to 24 h were recorded. Litter weight was taken every 2 days from day 0 to day 12 of lactation (L12). On L12, dams and pups were separated at 0600. Pups were weighed and then immediately euthanized. After 3 h of separation, dams were milked, euthanized and blood and tissues collected (B).

method. Blood was collected from the dam via cardiac punction and placed in an 7.5% EDTA K3 plasma separating tube (Covidien, Dublin, Ireland). Dams were then weighed.

Note, time is expressed in a 24-hour format (e.g., 0600 indicates 6:00 AM). In this study, "day" refers to the period from 0600 to 1800, and "night" refers to 1800 to 0600, regardless of the light pattern in the PS group.

## Weigh-suckle-weigh to estimate milk production and litter parameters

On lactation d10, dam was removed from cage for 2 h between 0800-1000. Time is expressed in a 24-hour format (e.g., 0800 indicates 8:00 AM). The cage was placed on a heating pad to keep pups warm. At the end of 2 h separation, pups were weighed and then dam was returned to the cage and pups were observed suckling and then weighed. The difference in weight from prior to and then after suckling was used to estimate milk production. Suckling duration was not timed but was observed until completion of suckling bout to ensure that all pups had adequate access to nursing. Birth litter size, litter weight, time of day of birth and pup survival to 24 h were also evaluated.

## Milk composition analysis

On lactation day 12, dams and pups were separated for ~ 3 hours prior to milking the dams. Milk collection was performed using a method modified from DePeters and Hovey [32], under light anesthesia using 3% isoflurane gas at a rate of 1.0 L/minute oxygen. To facilitate milk let down, dams were administered an intraperitoneal injection 0.2 mL of oxytocin (IP 20 Units/mL, Vet One, Boise, ID USA). The nipple area was dampened with 18-ohm water. A rubber stopper was placed in a 2 mL microcentrifuge tube, through which two 21 G X 1.5 in

needles were inserted: one for suction to collect milk, and the other connected to tygon tubing linked to an electric pump (Swing Breast Pump, Medela, Baar, Switzerland). Following collection, milk was place in -80ºC until analysis.

Milk was thawed overnight at 4˚C under constant rotation to homogenize. Milk was diluted 1:100 with phosphate buffered saline (PBS) for analysis of protein content using a Bradford assay (Fisher Scientific, Waltham, MA, USA) and 1:500 with PBS to measure lactose using a commercial kit (MAK017, Sigma-Aldrich, St. Louis, MO, USA). Milk TG was measured by preparing a 1:5 dilution using the Cayman Chemical kit (Ann Arbor, MI, USA). TG, which comprise approximately 98% of milk fat, were measured as a representation of total fat content.

## Blood and plasma analyses

Glycated hemoglobin (HbA1c) is a validated marker of glycemic control in mice [33–35]. Whole blood HbA1c was measured in undiluted whole blood collected in EDTA tubes using a commercial assay kit (Crystal Chem, Elk Grove Village, IL, USA). Plasma triacylglycerol (TG) was measured following 1:5 dilution with PBS using the Cayman Chemical kit. Plasma prolactin was measured in a 1:20 dilution of plasma using an ELISA kit (ab100736, Abcam, Waltham, MA, USA). All assays were performed following the manufacturers' instructions.

## Total feed intake

Feed intake was measured for the period of the day (0600–1800) and night (1800–0600). Intake was determined by weighing feed at 0600 and 1800 daily. To calculate the amount of food consumed during the day, the amount (grams) of food the mice had at 1800 was subtracted from the amount of food they started with at 0600 (0600 - 1800). For the night value, day weight was subtracted from the night weight (1800 - 0600). Values were recorded then assigned to physiological state, with early pregnancy encompassing d1-6, mid pregnancy d7-13 and late pregnancy d14-parturition, day of parturition, and then early lactation d1-4, mid lactation d5-8 and late lactation d9-12. Kcal of intake was determined by multiplying grams consumed by 3.85 kcal/g for CON, or 5.24 kcal/g for HF.

## Fecal collection, extraction and analysis of corticosterone content

Feces were collected twice daily from the cages to capture day (0600-1800) and night (1800-0600) output in early pregnancy (EP) d5-d7, mid pregnancy d7-d13, late pregnancy d14 to parturition (~d16), and lactation from day 4 to day 6. Mice were transferred to a new cage with fresh corn cob bedding and fecal material was separated from the bedding, collected, and stored at −20˚C.

To extract corticosterone (pg.mL$^{-1}$), feces were aliquoted to ~ 0.2g and then ground using a Precellys 24 Lysis & Homogenization (Bertin Technologies, Montigny-le-Bretonneux, France). After grinding, 2 mL/0.2 g of ethanol was added, and the sample was centrifuged. Approximately 1.3mL of the supernatant was pipetted into 2 separate tubes and dried in the speed vac for an hour. Once samples were dried, they were stored at -20˚C. Samples were reconstituted in 32.5 µL of absolute ethanol and analyzed using the corticosterone Enzyme Linked Immunosorbent Assay (ELISA) from Arbor Assays (catalog no. K014, Ann Arbor, MI, USA) following manufacturer's protocol. Absorbance was read at 450 nm on the Spark 10M multimode microplate reader (Tecan Trading AG, Switzerland).

## Lipid profiling analysis

Lipid extraction was performed according to the Bligh & Dyer protocol [36]. Briefly, 10 µL of milk and 100 µL of plasma were used for extraction, and deionized water (DI H$_2$O) was added in a microtube to bring the final volume to 200 µL. Tissue samples were individually weighed

and ~126 mg of liver and ~156 mg of mammary gland was homogenized in 2 mL vials with 1.4 mm ceramic (zirconium oxide) beads with 500 μL of water using the Precellys24 tissue homogenizer (Bertin Technologies, Rockville, MD, USA). Part of the homogenate (200 μL) was transferred to a microtube. Then, for all the matrices (liver, mammary gland, milk and plasma), 550 μL of methanol and 250 μL of chloroform ($CHCl_3$ + 0.01% BHT) were added and vortexed for 10 seconds to form a one-phase solution. The samples were incubated for 15 min at 4 °C. Next, another 250 μL of ultrapure $H_2O$ and 250 μL of $CHCl_3$ + 0.01% BHT were added. Samples were then centrifuged at 15,000 g for 10 min to enhance polar from non-polar phase separation (two-phase solution). The organic phase was transferred to a new tube. Additionally, an aliquot of the lipid extract of each sample was mixed by treatment to produce pooled samples. All tubes were dried in a centrifugal evaporator (Savant SpeedVac AES2010, ThermoFisher Scientific, San Jose, CA). Dried lipid extracts were stored at −80 °C until mass spectrometry analysis.

MRM profiling analysis was conducted in two phases, discovery and screening. For the discovery phase, samples pooled by treatment and tissue-matrix (16 pools) were profiled for a list of MRMs related to lipid species generated by combining the *m/z* for the molecular ion based on the LIPID MAPS® online database and a class- or fatty acyl loss-specific product ion [37]. To prepare such list, constitutional isomers were combined into one single entry in which only the lipid class, number of total carbons across fatty acyl residues and unsaturation number is listed. This strategy allowed compaction of the large number of entries (16,257) by one order of magnitude (3,437 MRMs in the final list) due to the large number of isobaric lipids. The 3,437 MRMs were profiled in the discovery step for the pooled samples of each experimental group by tissue-matrix. Of the 3,437 MRM profiled in the discovery phase, 1,491 were found to be at least 1.3-fold higher than the blank in at least one pooled sample. These 1,491 MRM were then used to interrogate individual samples in the screening phase of the MRM profiling analysis.

Due to the limited time of signal provided by the sample at flow injection, the 1,491 MRMs profiled in the screening phase, were organized into 10 methods (Supplementary S1 File), each one involving 2 minutes of data acquisition for a maximum of 250 MRM transitions. Dried lipid extracts were reconstituted in acetronitrile (ACN) + methanol (MeOH) + 300 mM ammonium acetate ($NH_4Ac$) 3:6.65:0.35 (v/v), with the same volume dried, which was 300 μL for pools, and 200 μL for each sample, composing the stock solution. The reconstituted extracts were individually diluted 100X for milk, 400X for mammary, 600X for liver and 250X for plasma, with injection solvent ACN + MeOH + 300 mM $NH_4Ac$ 3:6.65:0.35 (v/v). The screening phase of the MRM profiling was conducted by injecting 8 μL of the diluted lipid extract to the ESI source of an Agilent 6410 QQQ mass spectrometer (Agilent Technologies, Santa Clara, CA, USA) using a micro-autosampler (G1377A). The precursor ion selection window was 0.7 Th. The capillary pump connected to the autosampler operated with a flow of 10 μL/min and a pressure of 150 bar. Capillary voltage on the instrument was 3.5-5 kV and the gas flow was 5.1 L/min at 300°C.

The ion intensities of all MRMs were recovered, and each lipid ion was normalized by the total ion intensity of each sample (normalization by SUM) within each lipid class (phosphatidylcholines (PC), phosphatidylethanolamines (PE), phosphatidylserines (PS), phosphatidylinositol (PI), phosphatidylglycerol (PG), diacylglycerols (DG), triacylglycerols (TG), ceramides (Cer), sphingomyelins (SM), cholesteryl ester (CE), acyl-carnitines (AC), and free fatty acids (FFA), to acquire relative amounts. Subsequently, all data were consolidated into a unified file for each matrix (liver, mammary, milk, and plasma). Data analysis was performed using MetaboAnalyst 6.0 [38]. Following upload, data were normalized by auto-scaling and checked for normal distribution. Multivariate statistical tests included principal component analysis

(PCA), and linear model, and univariate tests included T-test, were used to identify lipids significantly influenced by diet and PS treatment.

### Data and statistical analysis

Statistical analyses, other than lipidome data, were performed using SAS 9.4 (SAS Inst. Inc., Cary, NC). The data were analyzed in a completely randomized design, and the animal was considered an experimental unit. The animals were randomly distributed in 1 of 2 dietary treatments and subsequently females were moved into one of the two experimental light rooms. The normality of the residues was verified by the Shapiro-Wilk test (UNIVARIATE procedure of SAS), and the Levene test compared the homogeneity of the variances. Variables with a continuous distribution such as feed intake, dam weight, birth litter size, weigh suckle weigh, TG, protein, lactose, prolactin and corticosterone concentration were analyzed using the MIXED procedure of SAS. When the time factor was not present, the statistical model included treatments (diet and light), stage (pregnancy and lactation), period of the day (day or night) as fixed effects, and mice as random effects. When the time factor was present (dam weight and litter weight), repeated measures models were fitted by multiple mixed linear models using the MIXED procedure of SAS, in which the statistical model included: the fixed effects of treatments (diet and light), period/day and their interaction, and the random effect of the animal. The experimental unit was the mice. The SLICE option using the LSMEANS/ PDIFF command was used to explore the interactions of data collection. The Kenward-Roger method was used to correct the degree of freedom of the denominator for the F test. The time of day of birth and pup death were evaluated by Chi-Square Test. The covariance structure was determined based on the lowest Akaike (AIC) information criteria value. Significance level was set at $P < 0.05$ and tendency towards significance at $0.05 < P < 0.10$.

## Results

### Feed intake and energy intake

The impact of HF diet and PS exposure on feed intake (amount in grams consumed) and energy intake (kcal consumed) by physiological stage (early, middle and late pregnancy and lactation) and period of the day (day: 0600-1800 or night: 1800-0600) were analyzed. Changes in amount of feed intake, particularly by time of day and physiological stage (i.e., early mid and late pregnancy and lactation) reflect changes in patterns of feeding behavior that may relate to dam diet, light treatment and the demands of physiological stage. Whereas changes in energy intake reflects the number of kilocalories consumed, which may reflect alterations in behavior in response to treatments or diets, as well as the metabolic needs based on the kilocalorie density of the diet.

Analysis of the number of grams consumed by mouse dams, found it significantly influenced by maternal diet and light exposure as well as period of the day (day or night), and physiological stage ($P < 0.05$, Table 1 in Supplementary S2 File). Energy intake was also significantly affected by PS exposure, time of day, physiological stage, and the interactions between PS exposure and period of the day, as well as the three-way interaction between stage, PS, and period of the day ($P < 0.05$, Table 1 in Supplementary S2 File). Overall, HF diet reduced grams of feed intake by dams relative to CON. Whereas PS exposure had an overall effect of increasing grams of feed intake and kilocalories of energy intake compared to the LD group (Table 1 in Supplementary S2 File). Analysis of the overall effect of physiological stage on feed intake founddams consumed more grams and kilocalories during lactation, with mid and late lactation showing higher levels of intake ($P < 0.01$) than all stages of gestation and early lactation. Overall, mice also consumed more food during the night period (1800-0600) compared to the

day (0600-1800) for both grams and kcal of food intake (Table 1 in Supplementary S2 File), although notably, the amount consumed during both the day and night increased progressively from early gestation to mid-lactation and intake.

There was also a significant interaction (P < 0.01, Table 1 in Supplementary S2 File) between chronic jet lag exposure and time of day for diet consumed in grams (Fig 2A) and kcal (Fig 2B). Post-hoc analysis revealed that both LD and PS mice consumed the most food during the night period. However, within the daytime period, PS mice had a higher feed and energy intake compared to LD mice (Fig 2A and 2B).

There was also a tendency for a three-way interaction between stage, light, and period of the day to influence feed intake (P = 0.05, Table 1 in Supplementary S2 File). In particular, PS mice consumed more feed during both the night and day in mid and late lactation compared to day or night in early, mid and late pregnancy (Fig 2C and 2D). Additionally, in the late stages of pregnancy and late lactation PS mice had a tendency for higher feed intake in the daytime hours(Fig 2C and 2D). Interestingly, mice exposed to PS had their second highest food intake during two specific periods: the nighttime period of early lactation and the daytime period of late lactation. The food intake was the same in both periods (Fig 2C and 2D).

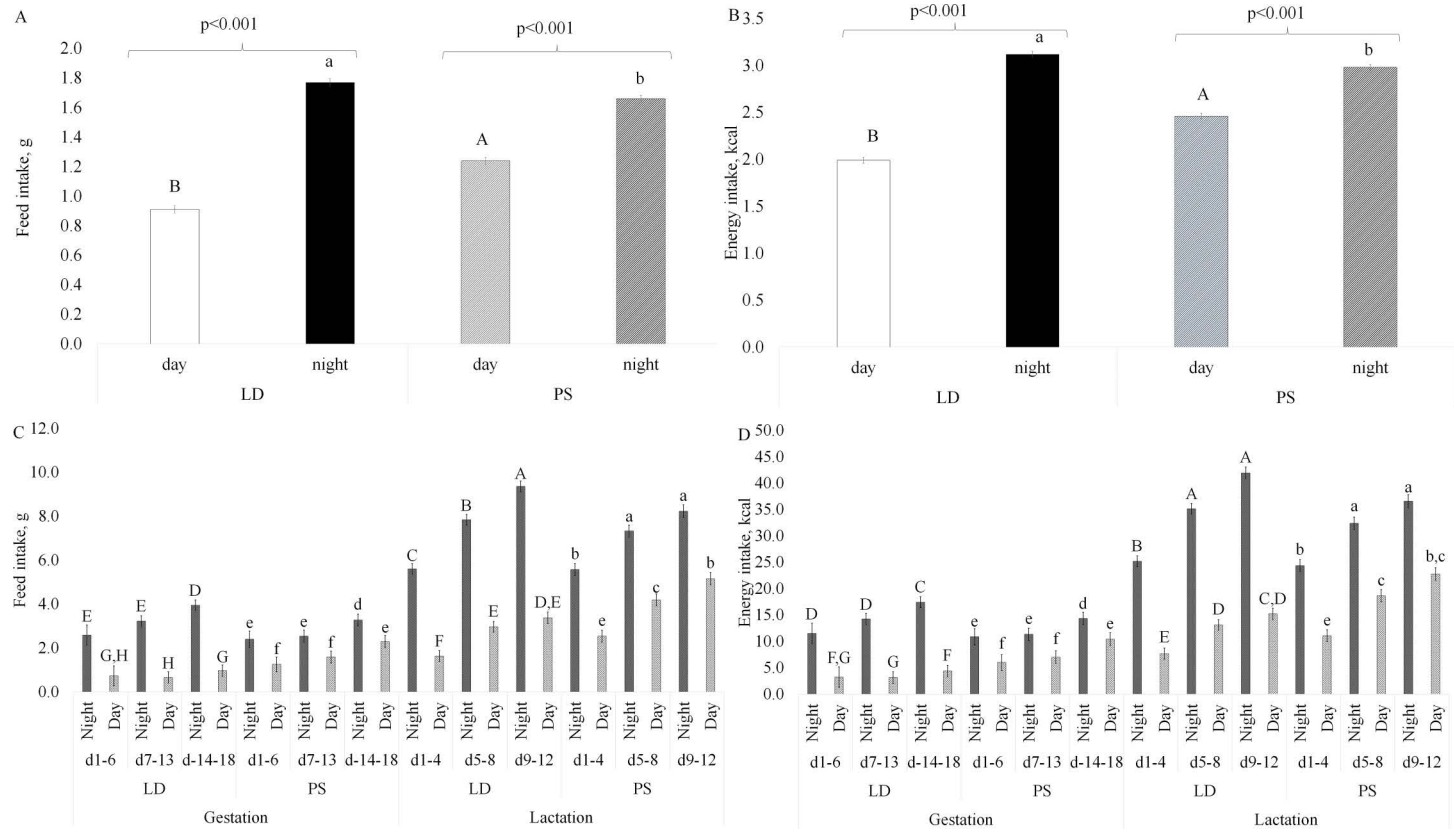

**Fig 2. Impact of light/dark (LD) exposure and phase shifts (PS) on diurnal feed and energy intake (A, B) and effect of light exposure, physiological stage, and time of day on feed and energy intake in mice (C, D).** LD, regular cycles of 12 hours of light and 12 hours of dark, and PS, mice exposed to 6-hour shifts in light-dark cycles every three days. Values are LS means ± SEM. Fig A and B: different letters indicate significant differences at P < 0.001, with uppercase indicating a difference between treatments during the day, and lowercase indicating a difference between treatments during the night. Fig C and D: different letters indicate a tendency (P = 0.05) for feed intake and a significant difference (P < 0.05) for energy intake, with uppercase indicating a difference between day and night for LD, and lowercase indicating a difference between day and night for PS treatment.

## Fecal corticosterone levels and fecal weight

Measuring fecal corticosterone provides a non-invasive means of evaluating changes of this stress related hormone in mice [21,39,40]. To assess the potential impact of HF diet and PS light exposure on fecal corticosterone levels by period of the day and physiological stage, feces were collected twice daily at 0600 and 1800 to represent night and day output in early pregnancy (EP) d5-d7, late pregnancy d14 to parturition (~d18), and day 4-6 of lactation. Although physiological stage had an overall effect on fecal corticosterone levels in mice (P = 0.0003, Table 2 in Supplementary S2 File), there was a significant interaction between light treatment exposure and physiological stage for corticosterone levels (P = 0.004, Fig 3A). Mice exposed to PS exhibited similar corticosterone levels across all physiological stages. This was in contrast to the LD group. In LD mice maternal corticosterone levels were highest in late pregnancy, followed by early pregnancy, and lowest during lactation (Fig 3A).

Total fecal weight produced over a 24 h period was greater in the PS group compared to those in the LD group (P = 0.001, Table 2 in Supplementary S2 File). Additionally, a

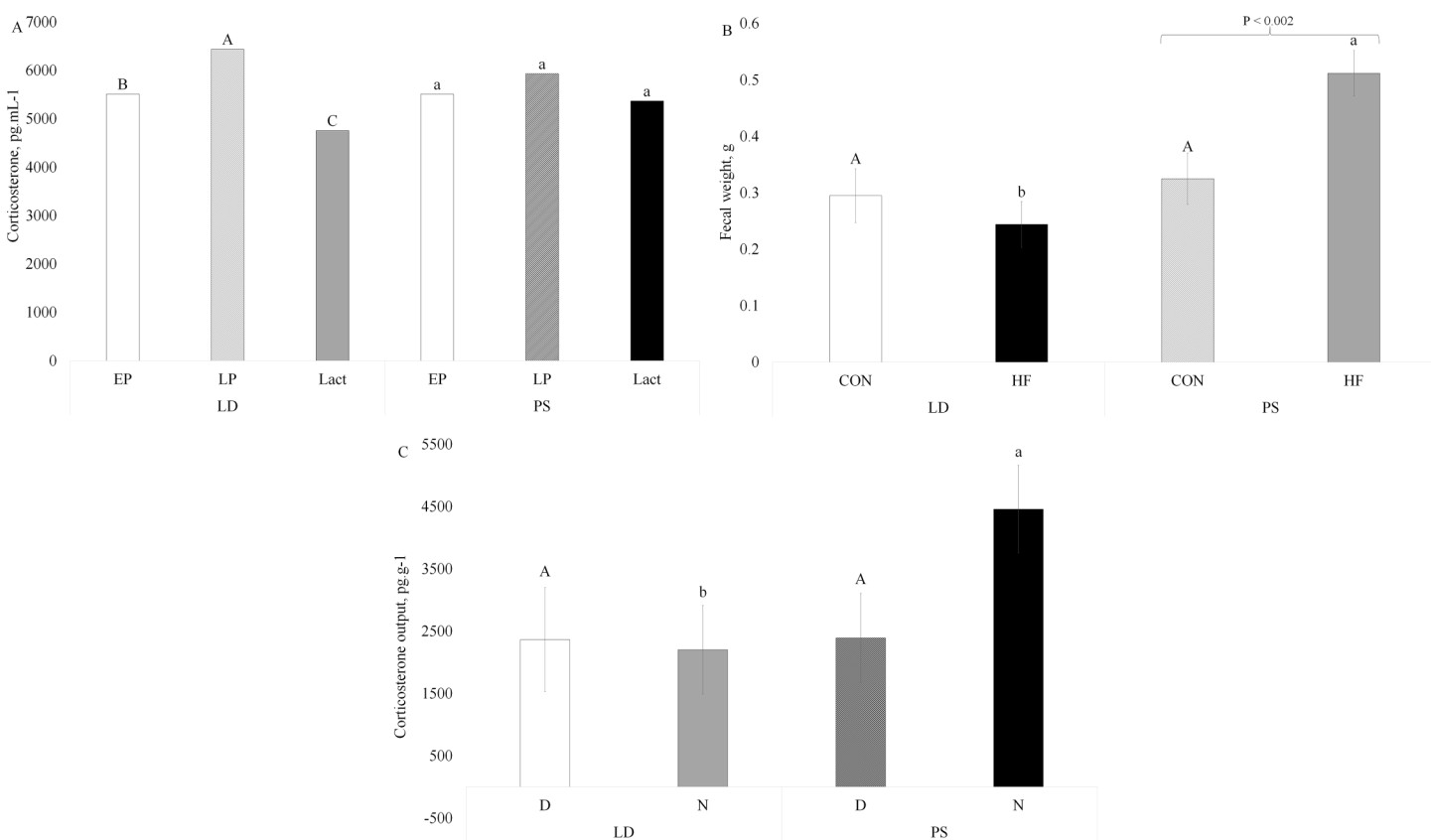

**Fig 3. Interaction between light and stage on fecal corticosterone levels (A), interaction between light exposure and diet on fecal weight (B) and interaction between light and day on corticosterone output in feces (C).** LD, regular cycles of 12 hours of light and 12 hours of dark, and PS, mice exposed to 6-hour shifts in light-dark cycles every three days. Control (CON) mice were fed a diet of 10% fat, and high-fat (HF) diet were fed a diet of 60% fat. Values are LS means ± SEM. Early pregnancy (EP) was from d5-d7, late pregnancy (LP) was from d14-d16, and lactation was d4-d6. Fig A: This interaction was significant, with a P-value of 0.004. Uppercase indicates a significant difference between stages within LD light, and lowercase indicates a significant difference between stages within PS light. Fig B: This interaction was significant, with a P-value of 0.008. Bracket indicates a significant difference (P < 0.002) between diets within lights. Uppercase indicates a significant difference between CON diet, and lowercase indicates a significant difference between HF diet across the lights. Fig C: This interaction tended to be significant, with a P-value of 0.068. Uppercase indicates a tendency to be a significant difference between day, and lowercase indicates a tendency to be a significant difference between night across the lights.

significant interaction between diet and light exposure was observed (P = 0.008, Fig 3B), with mice exposed to PS and fed a HF diet producing the greatest amount of feces by weight. The differences in fecal weight contributed to differences in corticosterone output levels, which were significantly affected by both light exposure and physiological stage (P < 0.05, Table 2 in Supplementary S2 File). Overall, corticosterone output levels were higher in the PS group compared to the LD group. Furthermore, there was a tendency (P = 0.068, Fig 3C) for an interaction between light exposure and period of the day, with PS treatment showing a near-significant increase in corticosterone levels at night compared to the day. Additionally, this trend suggests that corticosterone levels were higher at night under PS treatment compared to the LD treatment during the same period (Fig 3C).

## Dam weight, circulating prolactin, TG and HbA1c

The impact of HF diet on dam weight was assessed at the end of the prepregnancy period and on lactation day 12. Although overall dam weight was significantly increased by the HF diet (P = 0.041, Table 3 in Supplementary S2 File), post-hoc analysis indicated that this was more stage specific. Dams were heavier on lactation d12 compared to the pre-pregnancy period (P < 0.001, Table 3 in Supplementary S2 File), and a significant interaction between diet and stage was observed (P = 0.026, Fig 4). After 4 weeks of pre-pregnancy diet feeding, mice on the HF diet weighed more than those on a CON. However, by lactation d12 there was no difference in weight between dams on HF and CON diets.

PS treatment, which began with mating confirmation, had no effect on final maternal weight. Neither PS treatment, nor diet impacted circulating levels of prolactin, TG or HbA1c in dam's plasma on lactation day 12 (P > 0.05, Table 3 in Supplementary S2 File).

## Nutritional composition of milk

Dams were milked on lactation day 12 to determine the impact of HF diet and PS treatment on milk lactose, protein and TG content. Neither diet nor PS treatment impacted levels of TG

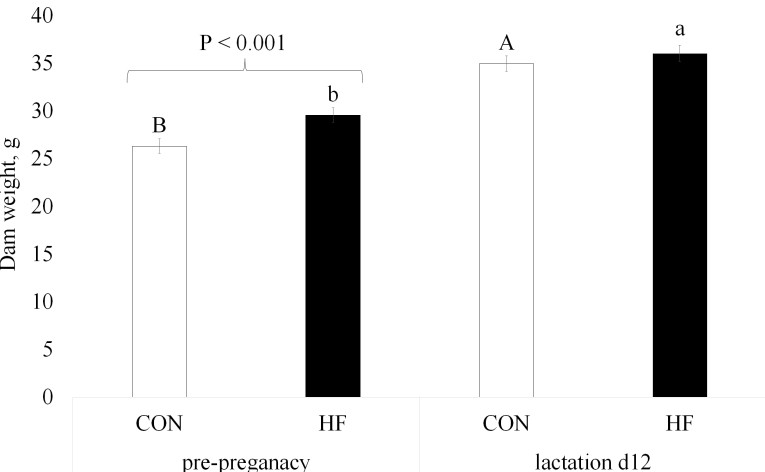

**Fig 4. Interaction of diet and physiological stage (pre-pregnancy and lactation d12) on dam's weight.** Control (CON) mice were fed a diet of 10% fat, and high-fat (HF) diet were fed a diet of 60% fat. Values are LS means ± SEM. Mice were fed respective diets *ad libitum* 4 weeks before mating. Neonates were euthanized on lactation day 12. Interaction diet*physiological stage had a P-value of 0.026. Brackets indicate significant difference at P < 0.001 between diets within pre-pregnancy. Uppercase indicates a difference between CON diet, and lowercase indicating a difference between HF diet across the stages.

and protein in milk (P > 0.05, Table 4 in Supplementary S2 File). Diet had an overall effect on lactose level, with the HF fed animals having greater milk lactose content (P = 0.007, Table 4 in Supplementary S2 File). A significant interaction between diet and light was observed for milk lactose levels (P = 0.037, Fig 5). In the CON diet group, mice treated with PS light had lower milk lactose levels compared to those in the LD treatment. However, in the HF diet group, chronic jet lag exposure had no effect on milk lactose levels. Across diets, there was no difference in milk lactose concentration for mice exposed to LD light. However, mice exposed to PS light had higher milk lactose levels when fed the HF diet than CON diet.

## Litter weight, birth litter size, weight suckle weight, time of day of birth and pup survival to 24 h

The impact of HF diet and PS treatment on birth litter size and time of day of birth was assessed by checking cages twice daily at 0600 and 1800 to determine if parturition occurred during the night (1800-0600) or day (1800-0600) and number of neonates born. Neonates were counted 24 h later, to determine survival, and then litters were standardized to 8 neonates per dam. Neither HF diet nor PS treatment affected birth litter size. Pup survival to 24 h was also not affected by diet nor chronic jet lag exposure (P > 0.05, Table 5 in Supplementary S2 File). There was a tendency (P = 0.056, Fig 6A) for chronic jet lag exposure to influence the time of day of birth, but diet had no effect on time of day of birth (P = 0.41, Fig 6B). In the LD group, 70.6% of births occurred during the day, whereas in the PS group, only 37.5% of births occurred during the day. The impact of HF diet and PS treatment on milk production levels was measured using weigh-suckle-weigh (WSW) on lactation day 10. There was no effect by maternal diet nor PS exposure on weight of milk produced (P > 0.05, Table 5 in Supplementary S2 File).

To assess the impact of diet and PS exposure on litter weight, litter weight was measured every 2 days. Litter weight was significantly affected by diet (P < 0.001, Table 5 in Supplementary S2 File). By lactation d12, the mean litter weight across all treatments was 74.25 g, compared to 13.81 g on the day of birth (0d) (P < 0.0001). This resulted in a significant interaction

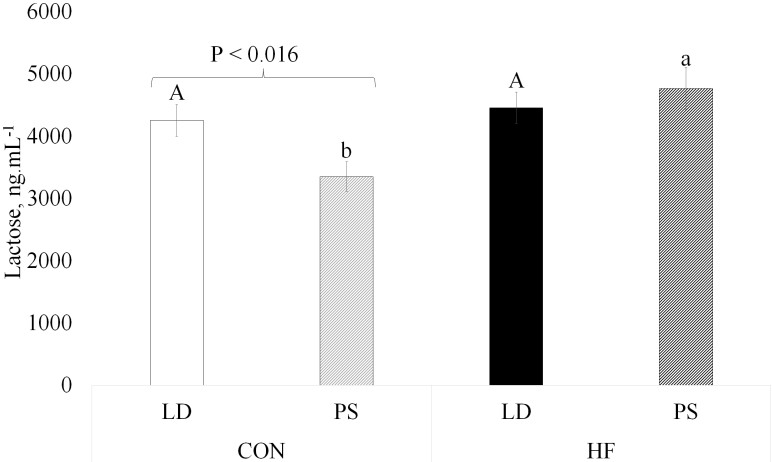

**Fig 5. Impact of diet and light on milk lactose.** LD, regular cycles of 12 hours of light and 12 hours of dark, and PS, mice exposed to 6-hour shifts in light-dark cycles every three days. Control (CON) mice were fed a diet of 10% fat, and high-fat (HF) diet were fed a diet of 60% fat. Values are LS means ± SEM. Interaction diet*light had a P-value of 0.037. Different letters indicate significant difference at P < 0.001, with uppercase indicating a difference between LD, and lowercase indicating a difference between PS.

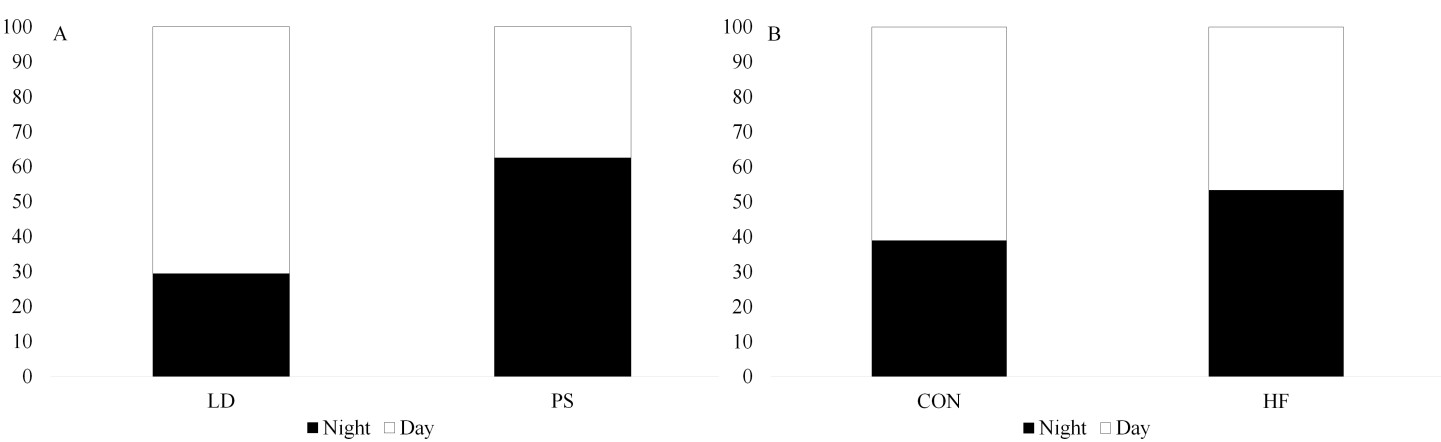

**Fig 6. Interaction between light and time of day (TOD) of birth (A) and interaction between diet and time of day (TOD) of birth (B).** LD, regular cycles of 12 hours of light and 12 hours of dark, and PS, mice exposed to 6-hour shifts in light-dark cycles every three days. Control (CON) mice were fed a diet of 10% fat, and high-fat (HF) diet were fed a diet of 60% fat. Values are LS means ± SEM. Significance was determined using a Chi-square test at a P < 0.05. Interaction light*time of day of birth had a P-value of 0.056 and diet*time had a P-value of 0.407. "Day" refers to the period from 0600 to 1800, and "night" refers to 1800 to 0600, regardless of the light pattern in the PS group.

(P < 0.0001) between diet and day with HF diet markedly increasing litter weight from day 8 to day 12 (Fig 7).

## Lipid profiles of liver, mammary, milk and plasma

Scores plots of principal component analysis (PCA) demonstrate that diet, but not chronic jet lag exposure, had a profound effect on lipid profiles of the liver, mammary gland, milk and plasma (Fig 8).

Linear model analysis of the overall effect of diet revealed that among 1204 lipids detected in the screening phase, diet altered 67.1% in milk, 58.1% in mammary gland, 27.2% in the liver, and 10.9% in plasma at P-adj < 0.05 (Table 1).

Linear model analysis of the overall effect of diet showed that diet significantly altered 67.1% of lipids in milk, 58.1% in mammary gland, 27.2% in liver, and 10.9% in plasma (P-adj

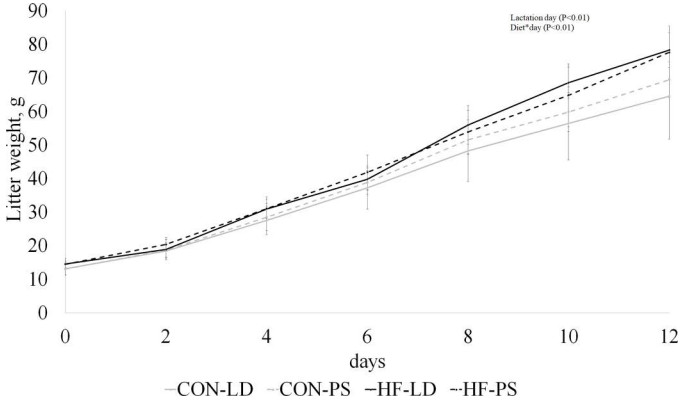

**Fig 7. Impact of phase shift (PS) and high-fat (HF) diet on litter weight.** LD, regular cycles of 12 hours of light and 12 hours of dark, and PS, mice exposed to 6-hour shifts in light-dark cycles every three days. Control (CON) mice were fed a diet of 10% fat, and high-fat (HF) diet were fed a diet of 60% fat. Values are LS means ± SEM.

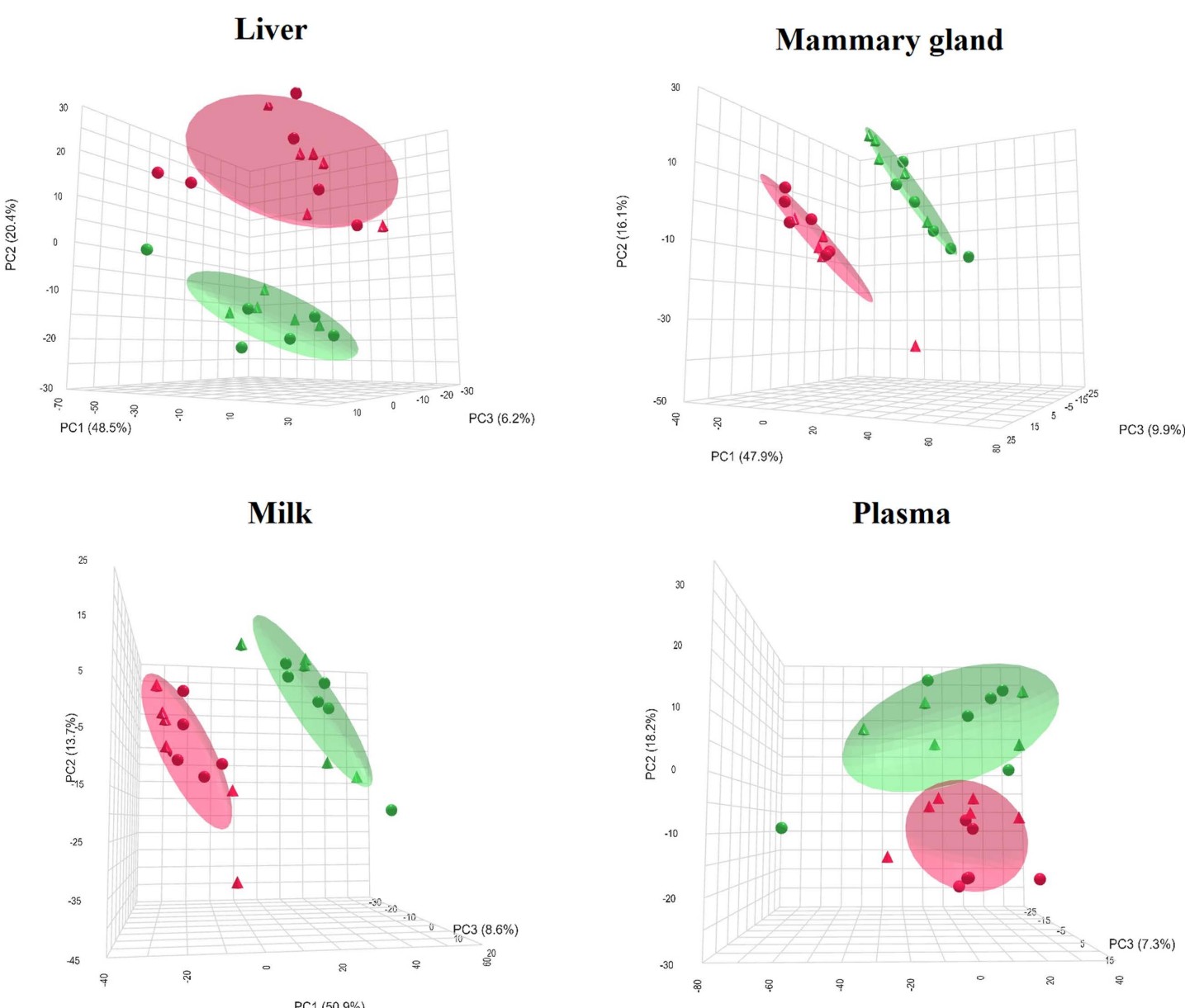

**Fig 8. PCA of lipid profiles in liver, mammary gland, milk, and plasma affected by diet but not by light.** Red and green represent CON and HF diet, respectively, while triangles and circles represent the PS and LD treatment, respectively.

**Table 1. Percentage of lipids affected by diet in liver, mammary gland, milk, and plasma based on linear model analysis.**

|  | Liver | | Mammary | | Milk | | Plasma | |
|---|---|---|---|---|---|---|---|---|
|  | n | % | n | % | n | % | n | % |
| Different at P-adj < 0.05 | 328 | **27.2** | 700 | **58.1** | 808 | **67.1** | 131 | **10.9** |
| Not different | 876 | 72.8 | 504 | 41.9 | 396 | 32.9 | 1073 | 89.1 |
| Total | 1204 | 100 | 1204 | 100 | 1204 | 100 | 1204 | 100 |

< 0.05). The color/gray shades represent the highest to the lowest percentage of differentially expressed lipids, indicating their proportion relative to the total lipids analyzed.

Across all tissues-matrices HF diet had the greatest impact on TG class of lipids, both increasing and decreasing abundance of individual species (Table 2). In liver, mammary and milk HF diet increased a relatively large proportion diacylglycerol (DG), whereas in the plasma the distribution of this class of lipids was relatively unchanged. In the liver the relative distribution of 15.05% of PC, 9.22% of PE and 7.28% PI lipids were decreased by HF diet. In the mammary gland 20.45% of PC lipids were decreased by HF, and 16.96% of PC and 15.90% of PE were decreased in milk. In plasma, 32.47% of PC lipid species were decreased (Table 2).

The impact of diet on total carbon length and number of unsaturated bonds across the fatty acyl groups within each lipid class and by tissue/matrix were analyzed to gain an understanding of how dams on a HF versus CON diet vary in these lipid characteristics. HF diet increased the percentage of DG in liver with total carbons across the two-fatty acyl ≤ 31 and between 37 to 40. HF diet also increased the percent of DG with higher degree of unsaturation compared to those in the CON group (Fig 9). Whereas dams on CON diets had a greater distribution of DG with 32 and 34 carbons and one unsaturated bond across the two fatty acyl groups.

In the mammary gland, mice fed HF diet exhibited a higher proportion of TG lipid class with longer carbon chains, ranging from 50 to 59, whereas those on a CON diet had a higher proportion of TG with ≤48 carbons. Additionally, HF diet increased the degree of unsaturation of TG lipids compared to mice on CON diets (Fig 9). HF diet also shifted distribution of DG in the mammary gland, with a greater proportion of 34 to 40 carbons, in contrast to the 37 to 44 carbon range observed in the CON diet group (Fig 9). Despite this alteration, the pattern of unsaturation of DG lipids in the mammary gland, remained largely similar to CON diet, although with a reduction in proportion and a slight increase noted for unsaturation of 3 to 5, 10, and 11.

Diet influenced milk TG total carbon number length across fatty acyl groups. Chains primarily ranged from 42 to 48 carbons in milk of mice fed CON diet, and in milk of mice on HF diet total carbon length were primarily greater than 50 (Fig 9). Diet also altered the distribution of milk TG by total number of unsaturated bonds, transitioning from a predominance of 0 to 4 unsaturation in the CON diet to 6 to 8 in milk of mice fed HF diet. Diacylglycerols comprised 34.48% of the total lipids altered in milk by HF diet (Table 2). Analysis revealed a U-shape distribution of DGs with a greater relative abundance in the HF group, with predominant degrees of unsaturation being 1, 2, 3, 8, and 9, and chain lengths ranging from 24 to 40 carbons (Fig 9). In contrast, DGs in CON groups displayed a different profile, characterized by a mix of saturated and unsaturated forms, with unsaturation levels primarily at 0, 1, 7, and 8, and longer chain lengths of 41 to 44 carbons.

Diet significantly impacted the TG composition in the plasma of mice, with those on the HF diet exhibiting longer carbon chains, ranging from 53 to 57 carbons across the three fatty acyl groups, and a higher prevalence of unsaturated bonds, with 6 to 8 double bonds (Fig 9). In contrast, TG of CON fed mice had shorter carbon chains, ranging from 48 to 53 carbons, and a different unsaturation profile, predominantly featuring 1, 2, and 9 double bonds. In plasma from mice fed the HF diet, a greater number of the total PC species profiled were significantly reduced compared to CON group. The PC species more abundant in plasma of HF mice had carbon chains ranging from 35 to 38 carbons and number of unsaturated bonds were 1, 2, and 4 across the two acyl molecules (Fig 9). In contrast, PC species more abundant in the CON group had carbon chains of 20 and 38 carbons, with 0 and 1 unsaturated bonds.

Linear regression analysis indicated no significant effect of chronic jet lag exposure on lipid profiles across all samples. Therefore, we tested whether within each diet lipid profiles differed between LD and PS treated mice using t-test. Differences in relative abundance of lipids

**Table 2. Distribution of lipid classes affected by diet in liver, mammary gland, milk, and plasma of mice fed with HF or CON diet according to linear model analysis.**

| | Liver | | | | Mammary gland | | | | Milk | | | | Plasma | | | |
| | HF>CON | | CON>HF | | HF>CON | | CON>HF | | HF>CON | | CON>HF | | HF>CON | | CON>HF | |
| | n | % | n | % | n | % | n | % | n | % | n | % | n | % | n | % |
|---|---|---|---|---|---|---|---|---|---|---|---|---|---|---|---|---|
| TG | 51 | **41.80** | 109 | **52.91** | 243 | **50.63** | 83 | **37.73** | 283 | **53.90** | 107 | **37.81** | 26 | **48.15** | 13 | **16.88** |
| CE | 3 | 2.46 | 2 | 0.97 | 2 | 0.42 | 2 | 0.91 | 2 | **0.38** | 17 | 6.01 | 0 | **0.00** | 4 | 5.19 |
| Cer | 4 | 3.28 | 1 | 0.49 | 4 | 0.83 | 2 | 0.91 | 1 | 0.19 | 0 | 0.00 | 0 | 0.00 | 1 | 1.30 |
| SM | 8 | 6.56 | 11 | 5.34 | 3 | **0.63** | 12 | 5.45 | 1 | **0.19** | 15 | 5.30 | 9 | 16.67 | 9 | **11.69** |
| DG | 39 | **31.97** | 12 | 5.83 | 165 | **34.38** | 22 | **10.00** | 181 | **34.48** | 14 | 4.95 | 0 | **0.00** | 4 | 5.19 |
| PC | 7 | **5.74** | 31 | **15.05** | 9 | **1.88** | 45 | **20.45** | 5 | **0.95** | 48 | **16.96** | 8 | **14.81** | 25 | **32.47** |
| PE | 5 | **4.10** | 19 | **9.22** | 6 | **1.25** | 17 | 7.73 | 2 | **0.38** | 45 | **15.90** | 8 | **14.81** | 6 | **7.79** |
| PI | 1 | **0.82** | 15 | **7.28** | 4 | **0.83** | 9 | 4.09 | 4 | **0.76** | 10 | 3.53 | 2 | **3.70** | 15 | 19.48 |
| PS | 2 | 1.64 | 5 | 2.43 | 1 | **0.21** | 13 | 5.91 | 3 | **0.57** | 14 | 4.95 | 1 | 1.85 | 0 | 0.00 |
| PG | 0 | 0.00 | 0 | 0.00 | 42 | **8.75** | 10 | 4.55 | 42 | **8.00** | 9 | 3.18 | 0 | 0.00 | 0 | 0.00 |
| AC | 0 | 0.00 | 0 | 0.00 | 1 | 0.21 | 2 | 0.91 | 1 | 0.19 | 3 | 1.06 | 0 | 0.00 | 0 | 0.00 |
| FFA | 2 | 1.64 | 1 | 0.49 | 0 | 0.00 | 3 | 1.36 | 0 | 0.00 | 1 | 0.35 | 0 | 0.00 | 0 | 0.00 |
| Total | 122 | 100 | 206 | 100 | 480 | 100 | 220 | 100 | 525 | 100 | 283 | 100 | 54 | 100 | 77 | 100 |

% Calculation: The percentage of lipids in each class was calculated by multiplying the number of lipids identified in each class by 100, then dividing by the total number of lipids detected. Bold percentages highlight the most affected lipid classes in each matrix. Full names of lipid categories include triacylglycerols (TG), ceramides (Cer), cholesteryl ester (CE), sphingomyelins (SM), diacylglycerols (DG), phosphatidylcholines (PC), phosphatidylinositol (PI), phosphatidylserines (PS), phosphatidylglycerol (PG), acyl-carnitines (AC), and free fatty acids (FFA). "HF>CON" indicates that lipids are increased in the HF group compared to the CON group. "CON>HF" indicates that lipids are increased in the CON group compared to the HF group.

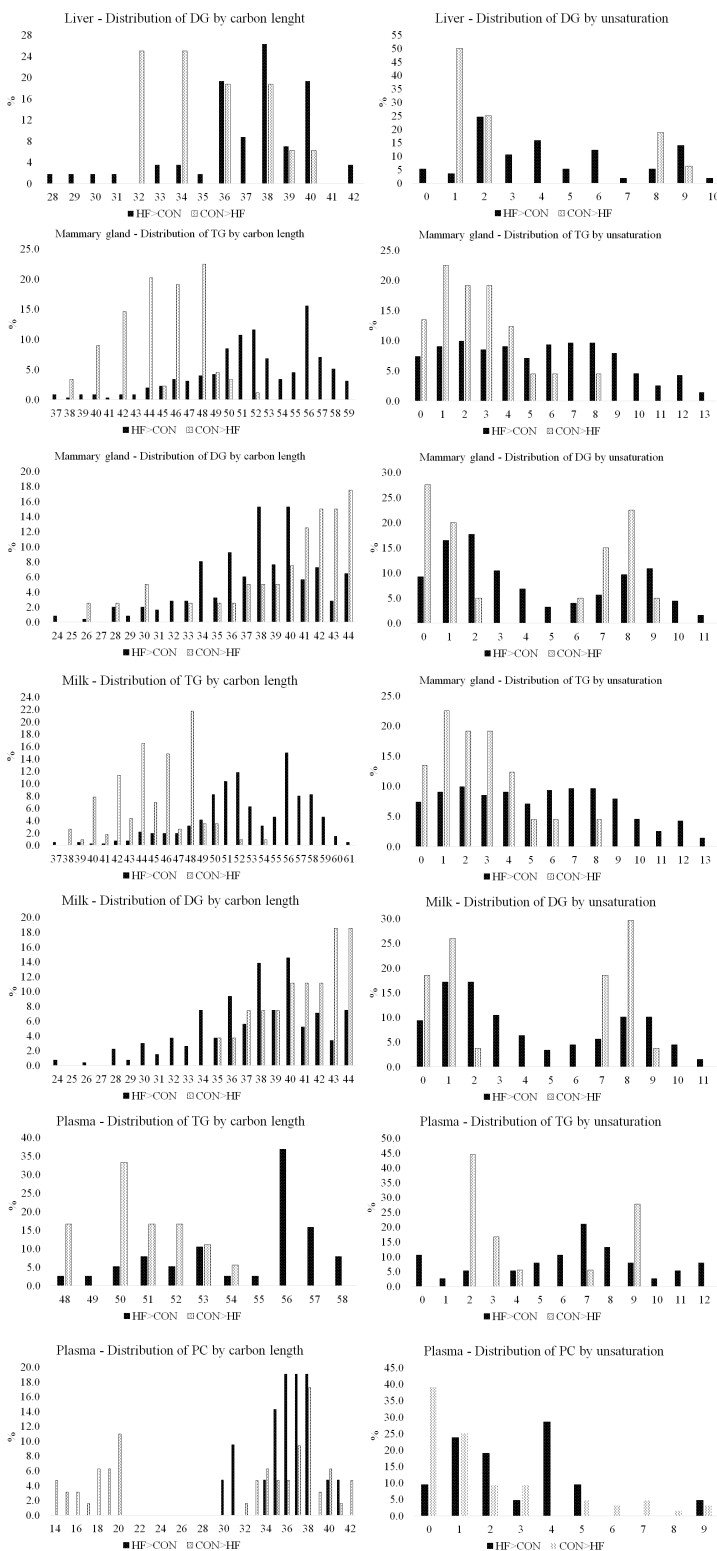

**Fig 9. Effects of HF diet on distribution of lipid classes in the liver, mammary gland, milk and plasma by total carbon chain length and degree of unsaturation.**

**Table 3. Comparison of significant lipid changes in various matrices under different LD and PS within HF and CON diets.**

| | CON LD-PS | | | | HF LD-PS | | | |
|---|---|---|---|---|---|---|---|---|
| | Liver | Mammary | Milk | Plasma | Liver | Mammary | Milk | Plasma |
| FDR P < 0.05 | 0 | 0 | 0 | 0 | 0 | 0 | 0 | 0 |
| Raw P < 0.05 | 35 (2.9%) | 33 (3.2%) | 6 (0.6%) | 67 (6.5%) | 11 (0.9%) | 120 (11.7%) | 9 (0.9%) | 19 (1.9%) |
| Total | 1204 (100%) | 1204 (100%) | 1204 (100%) | 1204 (100%) | 1204 (100%) | 1204 (100%) | 1204 (100%) | 1204 (100%) |

One-factor analysis, T-test with raw P-value 0 < 0.05. Significant lipid species are described in supplementary S3 File.

between LD and PS mice became apparent only after reducing the stringency to raw P-value < 0.05 (Table 3). Within CON diet, PS treatment significantly altered the abundance of 6.5%, 3.2%, 2.9%, and 0.6% of lipids profiled in plasma, mammary gland, liver and milk (Table 3). Within the HF diet, 11.7%, 1.9%, 0.9%, and 0.9% in mammary gland, plasma, liver and milk, respectively.

No patterns in lipids modified by chronic jet lag exposure within the CON diet group were observed for any tissue-matrices (Supplementary S4 File). PS treatment in the HF diet group altered the relative distribution of 120 lipids in mammary gland, of which 95% were decreased in abundance, whereas, in the CON diet group, PS treatment affected 33 lipids, with 78.8% showing increased abundance. Of the lipids reduced by PS treatment within HF diet, 45 were PC and 32 PE (Fig 10A). In contrast, these same lipid classes were increased by PS with CON diet, which 7 were PC and 12 were PE (Fig 10B).

The response to PS treatment was clearly influenced by the HF diet, as different lipid classes were downregulated in one diet but upregulated in the other, demonstrating light-diet interactions based on relative amount. To identify the consistent impact of PS treatment regardless of the diet, we analyzed lipids, from the most abundant classes, that were down-regulated in the HF-PS group and upregulated in the CON-PS group within the PC and PE classes using the Venny 2.1 tool [41]. This comparison explicitly shows the diet by light

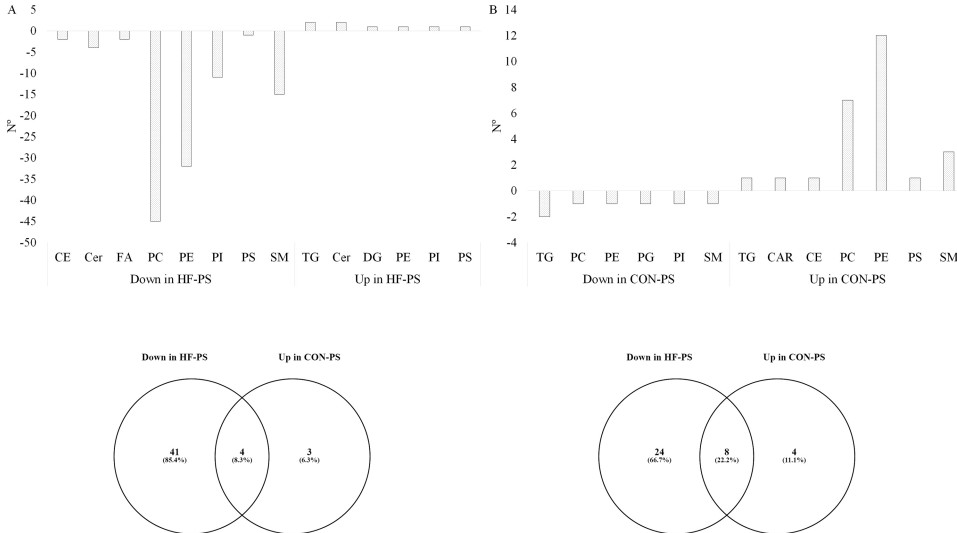

**Fig 10. Distribution of downregulated and upregulated lipids in mammary gland under PS within the HF or CON diet (A, B) and lipid classes downregulated in HF-PS and upregulated in CON-PS highlight the consistent effect of PS treatment across diets (C, D).**

**Table 4. Impact of PS treatment on lipid species and their isobars in the PC and PE classes across different diets.**

| Tentative attribution | Parent ion (*m/z*) | Product ion (*m/z*) |
|---|---|---|
| PC(14:0), LPC(15:0), LPC(O-16:0) | 482.4 | 184.1 |
| PC(33:3), PC(O-34:3), PC(P-34:2) | 742.6 | 184.1 |
| PC(42:4) | 866.7 | 184.1 |
| PC(44:5) | 892.7 | 184.1 |
| PE(33:2), PE(O-34:2), PE(P-34:1) | 702.5 | 561.5 |
| PE(37:3), PE(O-38:3), PE(P-38:2) | 756.6 | 615.6 |
| PE(39:6), PE(O-40:6), PE(P-40:5) | 778.6 | 637.6 |
| PE(41:6), PE(O-42:6) | 806.6 | 665.6 |
| PE(42:10), PE(41:3), PE(P-42:2) | 812.6 | 671.6 |
| PE(42:5) | 822.6 | 681.6 |
| PE(42:8), PE(41:1), PE(O-42:1), PE(P-42:0) | 816.7 | 675.7 |
| PE(44:4) | 852.6 | 711.6 |

Lipids shown in this table were downregulated in the HF-PS group and upregulated in the CON-PS group, with a focus on the PC and PE classes. These trends were identified using the Venny 2.1 tool and are also depicted in Fig 10.

interaction (Fig 10C and 10D). For the PC class, we found four lipid species and their isobars that were consistently affected by PS treatment, regardless of the diet (Fig 10C and Table 4). Similarly, for the PE class, we identified eight lipids and their isobars that were similar under both diets (Fig 10D and Table 4).

## Discussion

Maternal adaptations to pregnancy and lactation can be modified by the female's environment and nutrition, impacting the development of their offspring. Herein we described the impact of a circadian disrupting environment and high-fat diet on maternal physiology-lipidome, milk composition and neonate outcomes. Our primary findings include alteration in maternal feed intake, fecal corticosterone and milk lactose levels by both diet and circadian disrupting environments. Within the CON diet group, milk lactose levels of PS mice decreased, while mice on a HF diet produced milk with greater lactose content than those on a CON diet. Additionally, corticosterone levels were elevated for PS group across all physiological periods—early pregnancy, late pregnancy and lactation. HF diet increased litter weight during lactation and profoundly impacted the distribution of lipid profiles in dams' liver, plasma, mammary and milk, increasing abundance of both longer and shorter chains and altering levels of double bonds. The shifts in lipid profiles induced by HF diet likely reflect the greater dependence on lipids from diet to support energetic needs of dam and milk fat production. Whereas although PS treatment had no overall effect on lipid profiles, there was diet by chronic jet lag exposure interaction on the PC and PE lipids in the mammary gland. The PS treatment tended to increase the number of births during the night hours. Moreover, HF diet consumption significantly decreased overall feed intake and increased fecal weight.

The reduction in feed intake by the HF diet was expected and agrees with our previous study that found the same changes in eating behavior [42], indicating less of a need of mice to eat to support pregnancy and lactation when given an energy dense diet. The HF diet increased lactose levels in the milk of dams. This finding is consistent with our previous report [43] and may reflect a more concentrated milk (lower percent of water). However,

lack of an effect of diet on milk protein or TG concentration does not support this interpretation. Alternatively, the HF diet may have provided greater substrates for lactose synthesis. The HF diet supplies more glycerol, a primary substrate for gluconeogenesis. In the mammary gland, glucose derived from gluconeogenesis is converted to galactose, which is then used for lactose synthesis [43–46]. Glycerol may have also been sourced from the greater energy stores of the HF mice, which weighed more than CON mice at the onset of pregnancy. These HF mice appeared to mobilize their fat stores to support energetic needs throughout the experiment, as evident by the lack of weight difference between the diet groups at the end of the study [42].

In addition to the overall effect of HF diet on elevating milk lactose, the significant diet by light interaction indicated that within the CON group the PS treatment significantly reduced milk lactose concentration. Circadian rhythms play a role in regulating milk production, and our group has previously shown that continuous light exposure, which disrupts circadian clocks, decreased lactose synthesis [42]. Our previous studies also showed a circadian rhythm of LALBA expression, a key component of the lactose synthase enzyme [47]. Work of others demonstrated circadian rhythms of lactose synthesis [48–50]. Therefore, circadian disrupting environments may depress milk lactose content. If these findings translate to humans, decreased lactose content of breast milk may negatively impact infants, as the lactose content of milk functions to promote neurological development of neonates and establishment of a healthy microbiome [51].

The higher fecal weight observed in the interaction of the HF diet and PS treatment may be due to the increased feed intake caused by circadian disruption. Mice in the PS group exhibited higher feed intake during the daytime, indicating a shift in feeding patterns due to circadian rhythm disruption. Although weight gain due to disrupted feeding patterns in the PS group was anticipated, it was only observed in the HF group. Similar to our previous study [42], the dams on the HF diet gained more weight, in the pre-pregnancy phase. The high caloric density of the HF diet led to increased body weight and fat accumulation. Additionally, hormonal changes involving corticosterone, leptin and insulin can influence energy balance and promote weight gain [52,53]. Obesity in females can negatively impact fertility by disrupting hormonal balance, impairing ovulation, and altering reproductive function leading to suboptimal outcomes [54–56]. Although, there was no effect of diet on fecal corticosterone concentration or total output, our previous studies found the HF diet increased fecal corticosterone output and hair corticosterone levels, both at the end of the pre-pregnancy period and on lactation day 12 [31,57].

While we did not observe changes in milk production levels at the single weigh-suckle-weigh experiment on lactation day 10, the increased lactose concentration in milk of the HF group may have contributed to the higher litter weight, as lactose serves as a vital energy source for neonates supplying a substantial portion of the calories required for growth and development [58]. Although not different in this study, our previous study found a higher percent fat in milk of HF fed dams [43]. The HF milk as a calorie-dense food has the potential to increase growth rate of litters as well as provide bioactive lipids that support and optimize litter development [59].

The PS treatment disrupted the normal corticosterone fluctuations associated with the physiological transitions of the dam from early to late pregnancy and into lactation. Mice exposed to a 12-hour LD cycle showed changes in fecal corticosterone concentration between these stages, reflecting the hormonal adjustments necessary for impending parturition [60]. During pregnancy, elevated levels of estrogen and progesterone influence the hypothalamic-pituitary-adrenal (HPA) axis, resulting in an increase of corticosterone production [60–62]. Typically, there is a suppression of the HPA axis during lactation period to mitigate stress

responses that could interfere with maternal behaviors, thereby protecting the offspring from elevated stress hormone levels in the milk and leading to lower corticosterone levels [63–67]. Such changes are essential for the healthy development of the offspring [68]. However, dams exposed to constant shifts in the photoperiod due to PS treatment exhibited persistently high corticosterone concentrations, which remained unchanged across the reproductive phases. This pattern potentially indicates a chronic stress response, as constant circadian or ultradian rhythm disruption can dysregulate the HPA axis. Consequently, this dysregulation results in sustained elevated corticosterone levels, through elevated glucocorticoids, which can impair the negative feedback mechanisms that typically regulate the HPA axis or increase adrenal sensitivity to adrenocorticotropic hormone (ACTH) [69–71]. Such alterations in corticosterone rhythms may disrupt metabolic processes, increase the risk of metabolic disorders, suppress immune function, and negatively impact various aspects of reproduction and maternal care [69,70,72]. Additionally, these disruptions can have long-term effects on offspring development and behavior [31,42,72]. A study of mice found antenatal treatment of exogenous glucocorticoids out-of-phase with maternal endogenous circadian rhythm increased their offspring anxiety as well as impaired their stress coping stress-axis regulation [73]. In humans, preterm infants born to women who received antenatal glucocorticoids in the evening had lower stress compensatory capacity at 5 years of age than those born to women who received treatments within the endogenous circadian rhythm of glucocorticoids [73].

Exploratory analysis of organ (liver and mammary) and matrix (plasma and milk) lipid profiles indicated that HF diet significantly altered lipid content. Maternal intake of HF diet increased the number of carbons in TG found in the plasma, mammary gland, and milk, as well as in the DG of the liver and PC of the plasma. In contrast, animals on CON diet had a greater prevalence of DG with a higher number of carbons across fatty acyl group in the mammary gland and milk. These results suggest that the HF diet suppress *de novo* lipogenesis, while simultaneously increasing desaturation, which alter the lipid profile by enhancing the incorporation of fatty acids into different types of lipids in the body, thereby influencing the overall lipid composition and metabolism [74]. Although the expected outcome is the suppression of both *de novo* lipogenesis and desaturation, and maintains or even increases the elongation of fatty acids [75], the observed increase in desaturation may occur in response to the abundance of dietary fat, which provides more substrates for desaturation enzymes.

Relative to animals on CON diet, HF diet increased fatty acid (FA) length in the TG lipid class but decreased length in the DG lipid class in the mammary gland and milk. The mammary gland primarily synthesizes fatty acids *de novo* that are less than or equal to 16 carbons, resulting in shorter FA chains incorporated into TG due to the high affinity of diacylglycerol acyltransferase 1 (DGAT1) for these substrates [76,77]. DG are intermediates to TG and phospholipid synthesis [78], with the metabolic channeling in the mammary gland primarily directing longer-chain DGs toward phospholipid synthesis, essential for the milk fat globule membrane structure [76]. HF diet can lead to lipid remodeling in mammary gland, including elongation and desaturation of fatty acids (FA) through enzymes such as stearoyl-CoA desaturase (SCD1) and elongases [79,80]. The elongation of C16 fatty acids to very long-chain fatty acids (C18 or longer) occurs through ELOVL-mediated chain elongation [81,82]. Other studies have shown that the HF diet induces ELOVL expression [82] or enhances the elongation flux from C16 to C18:1 [83]. These changes in FA distribution in TG and DG in response to HF diet, as indicated by total carbon length and unsaturation, likely reflects the metabolic intermediates and final products.

Changing the FA chain lengths in TG and DG in milk can have significant consequences for neonates, potentially predisposing them to metabolic disorders or obesity later in life [84]. Longer-chain FA in TG increase the energy density of milk, leading to higher caloric

intake, faster weight gain, and potentially long-term effects on body composition and metabolic health [85,86]. In contrast, shorter-chain FA in DG are more easily digested, whereas longer-chain FA in TG reduce fat absorption efficiency, potentially causing gastrointestinal discomfort or malabsorption issues [87]. Neonates, especially those with fatty acid oxidation disorders, might experience an increased risk of metabolic decompensation and accumulation of toxic metabolites in mitochondria [88,89]. Additionally, longer-chain FA in TG reduce the availability of medium-chain FA for rapid ketogenesis, which can impact brain development, as ketones are vital for neonatal brains [90]. Alterations in FA chain lengths can also affect milk fat globule membrane structure, changing the bioavailability of membrane-associated nutrients and the stability of milk fat globules, which in turn affects digestion and absorption [91–93].

Circadian disruption as imposed by the chronic jet-lag model [94], impacts the molecular and physiological processes across the body causing a misalignment in the timing of biological functions that are usually synchronized with the light-dark cycle and profoundly affect metabolic homeostasis to include lipid and glucose metabolism [95–97]. In addition, disrupting the normal feeding-fasting cycle also affects lipid metabolism [98]. Our study revealed that lipid class PC and PE were impacted by the PS treatment, in a diet specific manner, reducing several species under HF diet and increasing under CON diet across all matrices. PC and PE are the major phospholipids in the membrane, comprising 45–55% and 15–25%, respectively. While PC is primarily associated with maintaining the integrity and functionality of the plasma membrane, PE plays a critical role within cytoplasmic lipid droplets and contributes to various metabolic processes [99]. Hepatic phospholipids, including PC and PE, decrease under HF diet conditions in mice with non-alcoholic fatty liver disease (NAFLD), a condition associated with high oxidative stress and liver damage [99]. In the mammary gland, the biosynthesis of PC and PE involves the CDP-choline (Kennedy) pathway and the phosphatidylethanolamine N-methyltransferase (PEMT) pathway [76,100]. Enzymes such as diacylglycerol choline/ethanolamine phosphotransferase (CEPT) and choline kinase α (CKα), all crucial in the Kennedy pathway, are influenced by diet, and hormonal changes during lactation, and possibly circadian rhythms [76,100,101]. Disruptions in these pathways, by circadian disrupting environments like the PS treatment, may influence the nutrient composition of milk, and impact pup growth and development [91,101,102].

Chronic jet-lag exposure affected four lipids in the PC class and eight in the PE class, regardless of the diet. Notably, some lipids contained odd-chain fatty acids (OCFA) and ether lipids. OCFA are less common in mammals and form through processes like the use of propionyl-CoA, or obtained through dietary sources [102] or alpha-oxidation of even-chain fatty acids [103]. The observed reduction in ether lipids, particularly those with longer carbon chains, suggests an increase in oxidative stress, as ether lipids play a crucial role in resisting lipid peroxidation and scavenging reactive oxygen species (ROS) [104,105]. Heightened oxidative stress can damage cells, negatively impacting milk production and quality, and potentially leading to impaired pup growth, weakened immune systems, and hindered cognitive development [106–109]. These findings imply that PS exposure might exacerbate oxidative stress in the mammary gland under HF diet conditions, highlighting the importance of lipid composition in maternal and neonatal health that may influence long-term health outcomes.

A limitation of this study is that the dams were weighed after the cardiac puncture, following the collection of approximately 1 mL of blood. This may have introduced variability in the recorded weights, as differences in blood volume collected could contribute to differences in the measurements. While the impact of this variability is likely small, it should be considered when interpreting the data. Also, while circadian disruption was induced through the chronic jet lag model, which is a well-established method for altering circadian rhythms [31,42,47], we

did not directly measure the impact on circadian rhythms (e.g., clock gene expression or loco-motor activity) in this study. This represents a limitation, as direct assessments of circadian function would provide stronger evidence of disruption.

## Conclusion

Our investigation explored the complex interaction between high-fat diet and exposure to circadian-disrupting light-dark phase shifts on various physiological parameters in pregnant and lactating mice that may contribute to the negative observations in offspring development in response to their exposure. High-fat diet and circadian disrupting PS treatment both altered feeding behavior. Dams subjected to HF showed reduced feed intake compared to CON group, while gaining more weight pre-pregnancy and showing no significant difference in weight at the end of lactation. PS increased feed intake without changes in dam or litter weight, suggests a decrease in metabolic efficiency. Maternal fecal corticosterone levels varied across early and late pregnancy and into lactation, however when dams were exposed to PS, corticosterone levels did not vary across physiological stage, remaining relatively elevated and indicating a greater exposure to stress throughout pregnancy and lactation. Lactose content of milk increased in response to HF diet, however in the CON group PS exposure decreased lactose content. Although birth litter size and pup survival were not affected by chronic jet lag exposure, there was a tendency for PS births to occur at night, supporting circadian system influences on the timing of parturition. HF diet profoundly impacted lipid profiles, with alterations supporting evidence for a greater dependence on dietary fats and their elongation and desaturation as well as oxidative damage across all tissues and matrices. Lipidome profiles also indicated PS exposure may increase oxidative stress damage due to reducing PC and PE lipid classes levels as well as potentially reducing their ether lipids. The changes in maternal variables likely reflect her health and metabolic status and affect the nutritional and developmental environment of offspring, and findings here may provide insight into the impact of maternal nutrition and environment on fetal and neonate development.

## Supporting information

**S1 File. MRMs profiled in the screening phase.**
(XLSX)

**S2 File. Supplementary tables.**
(DOCX)

**S3 File. Description of significant lipids changes in each matrix under LD and PS within HF and CON diets.**
(XLSX)

**S4 File. Lipids modified by chronic jet lag exposure within the CON diet group.**
(DOCX)

## Author contributions

**Conceptualization:** Theresa Casey.

**Data curation:** Kelsey Teeple, Sara Brook Scinto, Christina Ramires Ferreira.

**Formal analysis:** Leriana Garcia Reis, Michayla Dinn, Jenna Schoonmaker.

**Funding acquisition:** Leriana Garcia Reis.

**Investigation:** Kelsey Teeple, Theresa Casey.

**Methodology:** Christina Ramires Ferreira, Theresa Casey.

**Project administration:** Kelsey Teeple, Theresa Casey.

**Resources:** Kelsey Teeple.

**Supervision:** Kelsey Teeple, Theresa Casey.

**Visualization:** Leriana Garcia Reis, Michayla Dinn, Christina Ramires Ferreira, Theresa Casey.

**Writing – review & editing:** Kelsey Teeple, Michayla Dinn, Jenna Schoonmaker, Sara Brook Scinto, Christina Ramires Ferreira, Theresa Casey.

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
