## [Decision Letter · Decision Letter 0]

26 Nov 2024

PONE-D-24-38949The impact of circadian disruption and high-fat diet on ICR mice reproductive competence, fecal corticosterone and lipid profiles in mid-lactation: implications for maternal-neonate healthPLOS ONE

Dear Dr. Casey,

Thank you for submitting your manuscript to PLOS ONE. After careful consideration, we feel that it has merit but does not fully meet PLOS ONE’s publication criteria as it currently stands. Therefore, we invite you to submit a revised version of the manuscript that addresses the points raised during the review process.

We look forward to receiving your revised manuscript.

Kind regards,

Henrik Oster, Ph.D.

Academic Editor

PLOS ONE

Additional Editor Comments:

Dear authors,

your manuscriüt has been seen by two reviewers and while both acknowledge the relevance of the topic, they both also raise serious concerns regarding the conceptualization of the paper and the presentation of the data. I agree with their views. In a potential revision, please specifically address all issues regarding circadian disruption and oxidative state measures. Also consider revising the abstract and introduction, and carefully check logic and flow in the presentation of the results.

Reviewers' comments:

Reviewer's Responses to Questions

**Comments to the Author**

1. Is the manuscript technically sound, and do the data support the conclusions?

Reviewer #1: No

Reviewer #2: Yes

2. Has the statistical analysis been performed appropriately and rigorously? 

Reviewer #1: N/A

Reviewer #2: Yes

3. Have the authors made all data underlying the findings in their manuscript fully available?

Reviewer #1: Yes

Reviewer #2: Yes

4. Is the manuscript presented in an intelligible fashion and written in standard English?

Reviewer #1: No

Reviewer #2: Yes

5. Review Comments to the Author

Reviewer #1: Dear authors,

I have carefully reviewed the manuscript entitled "The impact of circadian disruption and high-fat diet on ICR mice reproductive competence, fecal corticosterone and lipid profiles in mid-lactation: implications for maternal-neonate health." The study addresses an interesting and important question—whether a high-fat diet (HFD) affects offspring development and milk production quality. However, several critical issues need to be addressed before the manuscript can be considered for publication in PLOS ONE.

While the topic is of significant interest, the complexity of the experimental design presents challenges in terms of presenting the findings clearly and comprehensively. A well-structured and concise presentation of the results and discussion sections is essential to effectively highlight the key findings. Unfortunately, the manuscript in its current form falls short of these expectations.

Major issues to address:

Abstract: The abstract is too long and reads more like a detailed list of experimental procedures and results, rather than a concise summary of the study's key objectives, findings, and conclusions. It should be revised to offer a clearer snapshot of the study’s main points. Additionally, the title is unnecessarily lengthy and should be shortened to better reflect the focus of the manuscript.

Introduction: The introduction places significant emphasis on circadian clock disruption, yet this aspect is not directly investigated in the study. For example, there are no measurements of clock gene expression or locomotor activity, which would provide evidence of circadian disruption. While the researchers induced internal clock disruption through chronic jetlag during gestation and lactation, this is not introduced or explained in the results section. If circadian disruption is not a core focus of the experimental work, I recommend revising the introduction to better align with the data presented in the manuscript.

Hypothesis on oxidative stress: The hypothesis regarding increased oxidative stress in chronically phase-shifted mice is not directly addressed in the experiments. If oxidative stress cannot be measured, the authors may need to reframe the hypothesis to more accurately reflect the collected data.

Experimental procedures: The experimental procedures, particularly the milking of the dams, require further clarification. While a sketch of the experimental setup was provided, the figures are blurry and difficult to interpret. Additionally, the text lacks clarity regarding how lactating females were selected for milking, when they were euthanised for blood collection, and how the timing of these procedures was managed. Further explanation is also needed for how the separate group of mice, milked regularly during the light phase, was handled. Clear and detailed descriptions of these procedures are crucial for the reader to understand the study design.

Pup handling and milk collection: The manuscript does not specify what happened to the pups after the dams were culled post-milking, nor does it indicate how much milk was collected from each female. If the dams were returned to their cages after milking, this could potentially reduce the amount of milk available to the pups, affecting their development. These details are important and should be addressed.

Results and discussion: Both the results and discussion sections require significant revision. The results are presented in isolated subchapters, lacking context for why specific experiments were conducted. This results in a disjointed presentation, making it difficult for the reader to follow the study’s overall narrative or draw coherent conclusions. A more integrated and cohesive structure is necessary. Furthermore, as previously mentioned, all the figures are blurry and difficult to interpret. Additionally, the discussion section needs to be significantly condensed.

Phrasing and terminology: Some phrasing in the manuscript is imprecise and requires revision for clarity. For example:

Instead of stating that "mice were treated with PS light" or "light treatment," it would be more accurate to describe the light manipulation as a "chronic jetlag protocol," where the mice were phase-shifted by 6 hours every 3 days during gestation and lactation.

Similarly, instead of stating "HF reduces food intake," it should be clarified that the diet "may lead to reduced food intake," as diets do not directly reduce food consumption but may influence it.

Reviewer #2: The manuscript is scientifically sound and of high interest. I believe the use of the PS treatment is a very interesting approach to circadian disruption. I have provided some comments in the manuscript and attach them to the review. I want to emphasize the importance of defining the terms day and night, since are not terms that adapt well to the PS treatment because it does not adapt to the natural 24-hour light/dark cycle.

6. PLOS authors have the option to publish the peer review history of their article (what does this mean? ). If published, this will include your full peer review and any attached files.

**Do you want your identity to be public for this peer review?** For information about this choice, including consent withdrawal, please see our Privacy Policy .

Reviewer #1: No

Reviewer #2: **Yes: ** Aridany Suarez-Trujillo

---

## [Author Response · Author response to Decision Letter 1]

18 Jan 2025

Response to Manuscript Reviewers: The impact of circadian disruption and high-fat diet on ICR mice reproductive competence, fecal corticosterone and lipid profiles in mid-lactation: implications for maternal-neonate health

We would like to thank you for the suggestions made, and the answers are listed below. In addition, the manuscript has been redrafted to make it clarifying, taking into account the suggestions made by the reviewers.

Reviewer #1:

1) I would just say "circadian disruption"

Response: The text was edited accordingly.

2) High-fat

Response: The text was edited accordingly.

3) Material and Methods: You used 20 and 20 in each group, but in the following sentence, you comment that the HF group had more animals to count with lower conception rates due to treatment.

Response: Apologies for oversight, yes this study assigned the same number of animals to each diet group.

4) Material and Methods: Could you add the conception rate for each group? I am assuming that you had 20 and 20 in CON and HD and 17 conceived in CON and 15 conceived in HF. Is that right?

Response: Yes, we have edited to correct, and give the pregnancy rates.

5) Material and Methods: I believe you should make this figure a Fig1A that is the colored boxes that explain the application of the treatments and Fig1B that contains the timeline. In addition, I think you should further develop the narrative in the figure caption.

Response: We have amended the figure, now A and B, with changes to the caption.

6) Material and Methods: This is not needed. The international agreement for time formatting is the 24-hour format, so you do not need to explain time formating.

Response: Thank you for your observation regarding the explanation of time formatting. We understand that the 24-hour format is internationally recognized. However, as this detail raised doubts for some of the co-authors during manuscript preparation, we chose to include the explanation to ensure clarity and avoid potential misunderstandings for all readers.

7) Material and methods: Did you weigh the dams after the cardiac puncture? how did you address the differences in weight due to variable amounts of blood collected in each animal. Please provide justification why the dams were weighed after blood collection.

Response: Thank you for bringing up this important point. We acknowledge that weighing the dams after blood collection introduces variability in the recorded weights due to the approximately 1 mL of blood collected during the cardiac puncture. This is a recognized limitation of our study. In future studies, we aim to address this limitation by standardizing the timing of weight measurements to occur prior to blood collection, ensuring more accurate and consistent data.

We added the information: “A limitation of this study is that the dams were weighed after the cardiac puncture, following the collection of approximately 1 mL of blood. This may have introduced variability in the recorded weights, as the blood volume collected could contribute to minor differences in the measurements. While the impact of this variability is likely small, it should be considered when interpreting the data. Future studies should address this limitation by standardizing the timing of weight measurements to occur prior to blood collection, ensuring more accurate and consistent data.”

8) Material and Methods: please explain how long were they allowed to suckle.

Response: Thank you for your question. We did not time the suckling events but instead observed until the completion of suckling. We added to a manuscript: “Suckling duration was not timed but was observed until completion to ensure that all pups had adequate access to nursing.”

9) Material and Methods: How were the milk samples preserved prior to the analyses?

Response: Milk samples were preserved at -80°C prior to analyses. We added to the manuscript: “Milk samples were stored at -80°C until analysis.”

10) Material and Methods: Please explain why did you measure TG and not total fat. I am aware that to measure total fat, the most common method is the crematocrit, which uses a large amount of sample compared with how much can be milked from a dam. However, the amount the milk fat is not entirely made up by TG. There is a small portion of other lipids.

Response: Thank you for your question. We measured TG because they account for approximately 98% of milk fat, providing a reliable representation of total fat content. We added to the manuscript: “TG, which comprise approximately 98% of milk fat, were measured as a representation of total fat content.”

11) Material and Methods: I does not matter where the homogenizer is located. I recommend removing it.

Response: The authors removed it.

12) Material and Methods: Explain what you mean by the variable stage. I am assuming that it is pregnancy vs lactation.

Response: Thank you for your comment. By "variable stage," we refer to the stages of pregnancy and lactation. The authors corrected the sentence: “When the time factor was not present, the statistical model included treatments (diet and light), stage (pregnancy and lactation), period of the day (day or night) as fixed effects, and mice as random effects.”

13) Results: Feed intake is measured in grams. What you are trying to refer here by saying "measured in kilocalories" is generally called Energy intake. I recommend you to switch to those two terms, feed intake when you refer at the amount of weight consumed and energy intake to the amount of energy consumed. Please, change throughout the manuscript, tables and figures.

Response: The authors verified the information and the terms were replaced throughout the manuscript, tables and figures.

14) Results: I think this sentence could be improved by switching the order. It is hard to follow since it has the "after 4 weeks of diet feeding pre-pregnancy"

Response: Thank you for the suggestion. We revised the sentence for clarity by reordering the phrasing: “Post hoc analysis indicated that after 4 weeks of pre-pregnancy diet feeding, mice fed a HF diet gained more weight than those on a CON diet.”

15) Results: Do you think "4 weeks of diet exposure during the pre-pregnancy period"

Response: We believe this was an incomplete thought, and the reviewer may be questioning whether 4 weeks of diet exposure was sufficient. While 4 weeks is a short period, we acknowledge that excessive obesity can reduce fertility, and the weight changes observed in this timeframe were sufficient to assess its potential impact on reproductive outcomes.

16) Results: Since the PS group had a constant shift of the lights, what do you mean by day? do you mean the interval 0600 to 1800? I recommend throughout the manuscript either explain that the time period 0600-1800 will be referred as "day" and the time period 1800-0600 will be referred as night, independently of the treatment and the actual light pattern happening for PS group.

Response: Thank you for the suggestion. We will clarify that "day" refers to the time period from 0600 to 1800 and "night" from 1800 to 0600, regardless of the light pattern in the PS group. We added to the manuscript: “In this study, "day" refers to the period from 0600 to 1800, and "night" refers to 1800 to 0600, regardless of the light pattern in the PS group.”

17) I would refer as litter weight. The term growth refers to the changes in weight by time, for example average daily gains used in livestock research.

Response: Thank you for the suggestion. We agree and changed to use "litter weight" instead of "growth" to more accurately reflect the measurement.

18) What does the color/gray shades on the differentially expressed lipids mean? I recommend to add an explanation on what the percentage mean. I understand they refer to the % of differentially expressed or not of the total.

Response: Thank you for the suggestion. The color/gray shades represent the highest to the lowest percentage of differentially expressed lipids, indicating their proportion relative to the total lipids analyzed. We added an explanation for clarity: “Linear model analysis of the overall effect of diet showed that diet significantly altered 67.1% of lipids in milk, 58.1% in mammary gland, 27.2% in liver, and 10.9% in plasma (P-adj < 0.05). The color/gray shades represent the highest to the lowest percentage of differentially expressed lipids, indicating their proportion relative to the total lipids analyzed.”

19) I am sorry for this comment because it is going to make this table even larger. I believe that the table has to stand by itself and be self-explanatory. I would add a legend with:

- how the % were calculated.

- What do the bold font on certain percentages mean?

- Full names for the lipid categories.

- What does HF>CON and others mean.

Response: Thank you for the helpful comments. We added a legend to the table to ensure it is more self-explanatory while keeping it concise: “% Calculation: The percentage of lipids in each class was calculated by multiplying the number of lipids identified in each class by 100, then dividing by the total number of lipids detected. Bold percentages highlight the most affected lipid classes in each matrix. Full names of lipid categories include triacylglycerols (TG), ceramides (Cer), cholesteryl ester (CE), sphingomyelins (SM), diacylglycerols (DG), phosphatidylcholines (PC), phosphatidylethanolamines (PE), phosphatidylinositol (PI), phosphatidylserines (PS), phosphatidylglycerol (PG), acyl-carnitines (AC), and free fatty acids (FFA). "HF>CON" indicates that lipids are increased in the HF group compared to the CON group. "CON>HF" indicates that lipids are increased in the CON group compared to the HF group.”

20) Results: Were these lipids over-expressed of under-expressed in PS versus LD?

Response: Thank you for the question. The lipids in the PS group were downregulated in the HF-PS group and upregulated in the CON-PS group, particularly in the PC and PE classes. This is reflected in Table 4 and Figure 10. We added as legend for table 4: “Lipids shown in this table were downregulated in the HF-PS group and upregulated in the CON-PS group, with a focus on the PC and PE classes. These trends were identified using the Venny 2.1 tool and are also depicted in Figure 10.”

21) Discussion: Mouse has a very marked nocturnal activity, do you think that the reason why the PS group had greater feed intake during 0600-1800 period than the LD group is simply because a larger percentage of the 0600-1800 periods matched with a dark phase in the holding facility, then the mice tended to consume more feed. What I am trying to grasp is if the mice consumed more because a direct effect of the PS trt on the mice physiology or because whatever the time in the day, the mice will always consume more feed in the dark phase.

Response: That's a potential explanation; however, mice need to eat when they are hungry. The increased feed intake during the 0600-1800 period in the PS group could reflect a direct physiological response to the phase-shifted lighting, rather than just a shift in the dark phase coinciding with the feed intake.

22) Discussion: This is what I was afraid when you measured TG in milk and not total fat. Cheng's article with HF vs CON diets, milk fat was measured with the crematocrit method instead of measuring TG. That can be the reason why you do not see differences in TG between treatments. Do you have any previous work that proves the correlation between milk TG and total fat %. I would recommend you to add it as a caveat or pitfall of the method used.

Response: Thank you for the suggestion. We acknowledge that TG represents only a component of total milk fat, accounting for 98%. We have added this as a weakness in the manuscript, noting that while TG is a reliable measure, it may not capture the full fat profile, which could explain the lack of differences between treatments.

23) Discussion: There's something wrong in the grammar of this sentence: "Elongation of FA (C16) to very-long-chain FA (C18 or longer) occurs via elongation of very long chain fatty acids (ELOVL)-mediated chain elongation [76,77].""

Response: Thank you for pointing that out. We corrected the grammar in the sentence: “The elongation of C16 fatty acids to very long-chain fatty acids (C18 or longer) occurs through ELOVL-mediated chain elongation [76,77].”

24) Discussion: Any cellular membrane?

Response: Thank you for pointing that out. We have clarified this in the manuscript with the following sentence: "While PC is primarily associated with maintaining the integrity and functionality of the plasma membrane, PE plays a critical role within cytoplasmic lipid droplets and contributes to various metabolic processes."

25) Discussion: I think the term night and day should be defined in here, the figure caption and the manuscript. What does night/day mean? is it based on the 0600-1800 and 1800-0600 periods or the dark or light phase that was happening in the LD or PS rooms at the time the births were observed. Same for feed/energy intake

Response: The information was added.

26) Figure 6: If this is a percentage, I so not think you should have 120%.

Response: The figure was corrected.

27) Figure 7: Is it there a light trt, diet or light trt*diet effect that is worth adding as p-values on one side of the figure. I would make these thicker, so it is easier to see.

Response: The information was added, and the changes were made to the figure.

Reviewer #2:

28) Abstract: The abstract is too long and reads more like a detailed list of experimental procedures and results, rather than a concise summary of the study's key objectives, findings, and conclusions. It should be revised to offer a clearer snapshot of the study’s main points. Additionally, the title is unnecessarily lengthy and should be shortened to better reflect the focus of the manuscript.

Response: The abstract was extensively edited, and titled changed.

29) The introduction places significant emphasis on circadian clock disruption, yet this aspect is not directly investigated in the study. For example, there are no measurements of clock gene expression or locomotor activity, which would provide evidence of circadian disruption. While the researchers induced internal clock disruption through chronic jetlag during gestation and lactation, this is not introduced or explained in the results section. If circadian disruption is not a core focus of the experimental work, I recommend revising the introduction to better align with the data presented in the manuscript.

Response: Thank you for your valuable comment. We acknowledge that the lack of direct measurements of circadian behavior, such as clock gene expression or locomotor activity, is a limitation of our study. However, the chronic jet lag model used in this study is a well-established method for inducing circadian disruption, and prior research supports its effectiveness in altering circadian rhythms, from the molecular to the behavioral level. We have added references to support use of this model for studying impacts of circadian disruption and added lack of measures on behavior and ‘rhythms’ themselves (although fecal output and feed intake may give some insight) as a limitation to the study in the discussion section.

That acknowledged and manuscript amended accordingly, measuring the impact of circadian disrupting environment on study outcomes was a major thrust of this work. In this 2X2 factor study, we report PS significantly impacted birthing time of day, lactose content of milk, corticosterone levels in feces, and levels of fecal corticosterone across the physiological-reproductive states. The fact it had minimal effects on lipidomes, surprised us, as we hypothesized it would, based on work of others, which demonstrated a central role of clocks in lipid metabolism in both liver and mammary, with plasma and milk lipid content regulated by these respective organs.

30) Hypothesis on oxidative stress: The hypothesis regarding increased oxidative stress in chronically phase-shifted mice is not directly addressed in the experiments. If oxidative stress cannot be measured, the authors may need to reframe the hypothesis to more accurately reflect the collected data."

Response: Thank you for your comment. You a

---

## [Decision Letter · Decision Letter 1]

6 Feb 2025

PONE-D-24-38949R1Exposure to circadian disrupting environment and high-fat diet during pregnancy and lactation alter reproductive competence and lipid profiles of liver, mammary, plasma and milk of ICR micePLOS ONE

Dear Dr. Casey,

Thank you for submitting your manuscript to PLOS ONE. After careful consideration, we feel that it has merit but does not fully meet PLOS ONE’s publication criteria as it currently stands. Therefore, we invite you to submit a revised version of the manuscript that addresses the points raised during the review process.

**- revise the abstract****- avoid long sentences****- revise results section with regard to readability and considering usability for a general audience**==============================

We look forward to receiving your revised manuscript.

Kind regards,

Henrik Oster, Ph.D.

Academic Editor

PLOS ONE

Journal Requirements:

Additional Editor Comments:

Dear authors,

while both reviewers acknowledge the effort you made in improving the manuscript, still concerns were raised regarding the presentation of data in the text. Please carefully address the comments of reviewer 1 regarding the abstract and the presentation of results, keeping in ind the general audience of this journal.

Reviewers' comments:

Reviewer's Responses to Questions

**Comments to the Author**

1. If the authors have adequately addressed your comments raised in a previous round of review and you feel that this manuscript is now acceptable for publication, you may indicate that here to bypass the “Comments to the Author” section, enter your conflict of interest statement in the “Confidential to Editor” section, and submit your "Accept" recommendation.

Reviewer #1: (No Response)

Reviewer #2: All comments have been addressed

2. Is the manuscript technically sound, and do the data support the conclusions?

Reviewer #1: Yes

Reviewer #2: Yes

3. Has the statistical analysis been performed appropriately and rigorously? 

Reviewer #1: Yes

Reviewer #2: Yes

4. Have the authors made all data underlying the findings in their manuscript fully available?

Reviewer #1: (No Response)

Reviewer #2: Yes

5. Is the manuscript presented in an intelligible fashion and written in standard English?

Reviewer #1: No

Reviewer #2: Yes

6. Review Comments to the Author

Reviewer #1: The authors conducted a substantial revision on their manuscript, however, the complexity of the project makes it still difficult to extract the main outcome of the project.

The abstract contains more methodological and result details than typically recommended for PLOS ONE. According to the journal's guidelines, it should provide a concise summary of the research objectives, methods, key findings, and conclusions. The study’s complex experimental procedures make conciseness challenging, the abstract currently includes excessive detail on diet types, pup survival and growth metrics, and various physiological measures (e.g., fecal corticosterone levels, milk composition, and lipid profiles). This reduces clarity and readability. Additionally, it exceeds the 300-word limit and includes statistical p-values, which should be omitted. Refining the abstract to focus on the study’s core findings while adhering to journal guidelines would improve its clarity and impact.

Results:

Lines 257–260 do not clearly indicate whether the peak kcal intake is from the PS group or the HFD group. The text should explicitly specify this, as it seems to refer to the PS group, but clarification is needed.

Although the authors have attempted to improve the structure in the results section, the text remains difficult to follow as the main thread is still unclear. The authors present interesting data, but they combine different life stages, such as pregnancy and lactation, along with various treatments, including chronic jetlag, HFD, and LD cycles, all at once. This makes the section challenging to read and follow.

I still believe that restructuring the results section would enhance clarity, particularly given the complexity of the experimental procedures. I suggest that the authors take a systematic approach, starting with food intake data during pregnancy (e.g., early vs. late stages) and comparing these between the PS and HF diets to determine whether these conditions affect energy intake. They could then proceed with food intake data specifically during lactation, highlighting that PS significantly impacts food intake, particularly in late lactation, whereas the HF diet did not result in increased food intake.

Also the text of glucocorticoid subchapter is difficult to follow because it contains long, complex sentences that present multiple ideas at once, making it hard to track the main case. The logical connections between different variables, such as physiological stage, light exposure, and diet, are not always clearly established, which can make it confusing to understand why specific comparisons are being made. The text also does not explicitly justify why certain conditions are being compared, which could leave the reader wondering about the reasoning behind the study design. A clearer structure with more concise sentences, explicit explanations of the rationale for comparisons, and a stronger emphasis on the main takeaways would significantly improve readability.

Minor changes:

Introduction:

74 change if oxidative change to of oxidative change

171 Feces" is a plural noun: feces were collected

Discussion:

Please include the study of Astiz et al. (doi: 10.1038/s41467-020-17429-5) in your discussion. This study demonstrates that maternal glucocorticoids influence offspring behavioral outcomes, which would be valuable to consider in the context of your findings on diet and chronic jetlag.

Reviewer #2: Thank you for addressing all the comments. I have reviewed your responses to the comments and the modifications in the manuscript and I believe the manuscript is now ready for publication.

7. PLOS authors have the option to publish the peer review history of their article (what does this mean? ). If published, this will include your full peer review and any attached files.

**Do you want your identity to be public for this peer review?** For information about this choice, including consent withdrawal, please see our Privacy Policy .

Reviewer #1: No

Reviewer #2: No

---

## [Author Response · Author response to Decision Letter 2]

17 Feb 2025

Response to review-2 PONE

We would like to thank reviewer 1 for their additional comments, and concerted efforts to improve the readability and clarity of our manuscript. We believe the changes made in the second revision have done so.

Reviewer #1: The authors conducted a substantial revision on their manuscript, however, the complexity of the project makes it still difficult to extract the main outcome of the project.

The abstract contains more methodological and result details than typically recommended for PLOS ONE. According to the journal's guidelines, it should provide a concise summary of the research objectives, methods, key findings, and conclusions. The study’s complex experimental procedures make conciseness challenging, the abstract currently includes excessive detail on diet types, pup survival and growth metrics, and various physiological measures (e.g., fecal corticosterone levels, milk composition, and lipid profiles). This reduces clarity and readability. Additionally, it exceeds the 300-word limit and includes statistical p-values, which should be omitted. Refining the abstract to focus on the study’s core findings while adhering to journal guidelines would improve its clarity and impact.

RE: abstract was shortened to 296 words, and better focused

Results:

Lines 257–260 do not clearly indicate whether the peak kcal intake is from the PS group or the HFD group. The text should explicitly specify this, as it seems to refer to the PS group, but clarification is needed.

RE: We thank you for encouraging us to address the readability-understandability of these data—and believe the changes made to this section do this. The same goes for comment below.

Although the authors have attempted to improve the structure in the results section, the text remains difficult to follow as the main thread is still unclear. The authors present interesting data, but they combine different life stages, such as pregnancy and lactation, along with various treatments, including chronic jetlag, HFD, and LD cycles, all at once. This makes the section challenging to read and follow.

RE: see above, we believe changes made to this section help better clarify outcomes.

I still believe that restructuring the results section would enhance clarity, particularly given the complexity of the experimental procedures. I suggest that the authors take a systematic approach, starting with food intake data during pregnancy (e.g., early vs. late stages) and comparing these between the PS and HF diets to determine whether these conditions affect energy intake. They could then proceed with food intake data specifically during lactation, highlighting that PS significantly impacts food intake, particularly in late lactation, whereas the HF diet did not result in increased food intake.

RE: as above

Also the text of glucocorticoid subchapter is difficult to follow because it contains long, complex sentences that present multiple ideas at once, making it hard to track the main case. The logical connections between different variables, such as physiological stage, light exposure, and diet, are not always clearly established, which can make it confusing to understand why specific comparisons are being made. The text also does not explicitly justify why certain conditions are being compared, which could leave the reader wondering about the reasoning behind the study design. A clearer structure with more concise sentences, explicit explanations of the rationale for comparisons, and a stronger emphasis on the main takeaways would significantly improve readability.

RE: Changes were made to this section to help clarify and highlight important outcomes.

Minor changes:

Introduction:

74 change if oxidative change to of oxidative change

RE: edited

171 Feces" is a plural noun: feces were collected

RE: edited

Discussion:

Please include the study of Astiz et al. (doi: 10.1038/s41467-020-17429-5) in your discussion. This study demonstrates that maternal glucocorticoids influence offspring behavioral outcomes, which would be valuable to consider in the context of your findings on diet and chronic jetlag.

RE: added reference and relevant findings to discussion

---

## [Editor Report · Decision Letter 2]

21 Feb 2025

Exposure to circadian disrupting environment and high-fat diet during pregnancy and lactation alter reproductive competence and lipid profiles of liver, mammary, plasma and milk of ICR mice

PONE-D-24-38949R2

Dear Dr. Casey,

We’re pleased to inform you that your manuscript has been judged scientifically suitable for publication and will be formally accepted for publication once it meets all outstanding technical requirements.

Kind regards,

Henrik Oster, Ph.D.

Academic Editor

PLOS ONE

Additional Editor Comments (optional):

Congrats on this nice story.
---

## [Editor Report · Acceptance letter]

PONE-D-24-38949R2

PLOS ONE

Dear Dr. Casey,

I'm pleased to inform you that your manuscript has been deemed suitable for publication in PLOS ONE. Congratulations! Your manuscript is now being handed over to our production team.

Kind regards,

on behalf of

Prof. Henrik Oster

Academic Editor

PLOS ONE
